# A Previously Undescribed Highly Prevalent Phage Identified in a Danish Enteric Virome Catalog

Lore Van Espen,[a] Emilie Glad Bak,[b] Leen Beller,[a] Lila Close,[a] Ward Deboutte,[a] Helene Bæk Juel,[b] Trine Nielsen,[b] Deniz Sinar,[a] Lander De Coninck,[a] Christine Frithioff-Bøjsøe,[b,c] Cilius Esmann Fonvig,[b,c] Suganya Jacobsen,[d,e] Maria Kjærgaard,[d,e] Maja Thiele,[d,e] Anthony Fullam,[f] Michael Kuhn,[f] Jens-Christian Holm,[b,c,g] Peer Bork,[f,h,i,j] Aleksander Krag,[d,e] Torben Hansen,[b] Manimozhiyan Arumugam,[b] Jelle Matthijnssens[a]

aKU Leuven, Department of Microbiology, Immunology, & Transplantation, Rega Institute, Division of Clinical & Epidemiological Virology, Laboratory of Viral Metagenomics, Leuven, Belgium

bThe Novo Nordisk Foundation Center for Basic Metabolic Research, Faculty of Health and Medical Sciences, University of Copenhagen, Copenhagen, Denmark

cThe Children's Obesity Clinic, accredited European Centre for Obesity Management, Department of Paediatrics, Copenhagen University Hospital Holbaek, Holbaek, Denmark

dDepartment of Gastroenterology and Hepatology, Centre for Liver Research, Odense University Hospital, Odense, Denmark

eDepartment of Clinical Research, University of Southern Denmark, Odense, Denmark

fStructural and Computational Biology Unit, European Molecular Biology Laboratory, Heidelberg, Germany

gFaculty of Health and Medical Sciences, University of Copenhagen, Copenhagen, Denmark

hMax Delbrück Centre for Molecular Medicine, Berlin, Germany

iYonsei Frontier Lab (YFL), Yonsei University, Seoul, South Korea

jDepartment of Bioinformatics, Biocenter, University of Würzburg, Würzburg, Germany

Lore Van Espen and Emilie Glad Bak contributed equally. Author order was decided by the corresponding authors.

**ABSTRACT** Gut viruses are important, yet often neglected, players in the complex human gut microbial ecosystem. Recently, the number of human gut virome studies has been increasing; however, we are still only scratching the surface of the immense viral diversity. In this study, 254 virus-enriched fecal metagenomes from 204 Danish subjects were used to generate the Danish Enteric Virome Catalog (DEVoC) containing 12,986 nonredundant viral scaffolds, of which the majority was previously undescribed, encoding 190,029 viral genes. The DEVoC was used to compare 91 healthy DEVoC gut viromes from children, adolescents, and adults that were used to create the DEVoC. Gut viromes of healthy Danish subjects were dominated by phages. While most phage genomes (PGs) only occurred in a single subject, indicating large virome individuality, 39 PGs were present in more than 10 healthy subjects. Among these 39 PGs, the prevalences of three PGs were associated with age. To further study the prevalence of these 39 prevalent PGs, 1,880 gut virome data sets of 27 studies from across the world were screened, revealing several age-, geography-, and disease-related prevalence patterns. Two PGs also showed a remarkably high prevalence worldwide—a crAss-like phage (20.6% prevalence), belonging to the tentative *AlphacrAssvirinae* subfamily, and a previously undescribed circular temperate phage infecting *Bacteroides dorei* (14.4% prevalence), called LoVEphage because it encodes lots of viral elements. Due to the LoVEphage's high prevalence and novelty, public data sets in which the LoVEphage was detected were *de novo* assembled, resulting in an additional 18 circular LoVEphage-like genomes (67.9 to 72.4 kb).

**IMPORTANCE** Through generation of the DEVoC, we added numerous previously uncharacterized viral genomes and genes to the ever-increasing worldwide pool of human gut viromes. The DEVoC, the largest human gut virome catalog generated from consistently processed fecal samples, facilitated the analysis of the 91 healthy Danish gut viromes. Characterizing the biggest cohort of healthy gut viromes from children, adolescents, and adults to date confirmed the previously established high

Address correspondence to Torben Hansen, torben.hansen@sund.ku.dk, Manimozhiyan Arumugam, arumugam@sund.ku.dk, or Jelle Matthijnssens, jelle.matthijnssens@kuleuven.be.

Previously undescribed, widely prevalent, Bacteroides-infecting temperate phage discovered during generation of a Danish enteric virome catalog.

interindividual variation in human gut viromes and demonstrated that the effect of age on the gut virome composition was limited to the prevalence of specific phage (groups). The identification of a previously undescribed prevalent phage illustrates the usefulness of developing virome catalogs, and we foresee that the DEVoC will benefit future analysis of the roles of gut viruses in human health and disease.

**KEYWORDS** human gut virome, virome catalog, healthy gut viromes, phages

Gut microbiota, consisting of bacteria, archaea, viruses, fungi, and other eukaryotic microorganisms, play a major role in human health and disease (1, 2). Both structural and functional imbalances of the gut bacteria, called dysbiosis, have been associated with diseases such as obesity (3), diabetes (4, 5), inflammatory bowel disease (IBD) (6), cancer (7), and neurological diseases (8). At the same time, research on human gut viruses, collectively called gut virobiota, is still in its infancy (9), although recent studies demonstrated associations with IBD (10, 11), diabetes (12, 13), liver disease (14, 15), and cancer (16).

Only a minority of the human gut virobiota consists of eukaryotic viruses, infecting human cells, fungal cells, and unicellular eukaryotes residing in the gut or infecting plant or animal cells transiting as part of the diet (17). The vast majority of viruses in the human gut are bacteriophages (phages), which rely on a bacterial host to reproduce (18). The close interplay between phages and bacteria, which are already implicated in numerous diseases, combined with the ability of gut viruses to directly interact with the human host (19, 20), led to gut viruses gaining more interest as potential disease biomarkers (16) and treatments for disease (21, 22). It is therefore important to shed more light on the virobiota, and their collective genomes referred to as the virome, as this will pave the way to unravelling complex interactions within the gut microbiota and their effect on the human host (23).

Recent progress in high-throughput sequencing technologies, viral enrichment procedures, and development of downstream viral bioinformatic tools has facilitated human gut virome studies investigating their association with health (24–29) or disease (30–34), as well as their dynamics (18). However, several significant challenges in studying human gut viromes remain (23). Most importantly, identification of viruses from metagenomes is hampered by incomplete databases (23) and therefore requires specialized viral identification tools, e.g., VirSorter (35), MetaPhinder (36), and DeepVirFinder (37), that do not (only) rely on similarity to known viral genomes/proteins but also look at genome structure to detect viral signatures. High viral mutation rates cause immense viral genetic diversity (38), thereby complicating viral identification based on homology to reference genomes. Even though a few virome databases recently emerged, they are often not gut-specific (IMG/VR [39], Reference Viral DataBase [RVDB; 40] and Earth's virome [41]) or focus only on phages or eukaryotic viruses (Gut Phage Database [GPD; 42], a "circular" phage database [43], and RVDB [40]). However, despite these developments, a large fraction of sequences originating from human gut virome studies cannot be identified as viral because they are not present in databases and are therefore called "viral dark matter" (23). Moreover, taxonomic characterization of human gut viruses is virtually impossible due to the major proportion of viruses being taxonomically unclassified (44, 45), despite ongoing efforts by the International Committee for the Taxonomy of Viruses (ICTV) (46). Thus, viral taxonomic analysis is mostly performed on scaffold level or on artificial taxonomic levels generated by gene-sharing tools, e.g., vConTACT2 (47) or GRAViTy (48). Finally, the lack of host information and functional annotation of proteins complicates the characterization of the phages and their interactions with bacteria in the gut (23, 49). The technical difficulties of identifying and characterizing viruses are numerous. Nevertheless, it is important to make progress in generating human gut virome catalogs and characterizing them to shed light on the viral dark matter. This was exemplified by the discovery of crAssphages from the cross-assembly of human gut metagenomes across publicly available data sets (50). This novel group of phages is now believed to be one of the

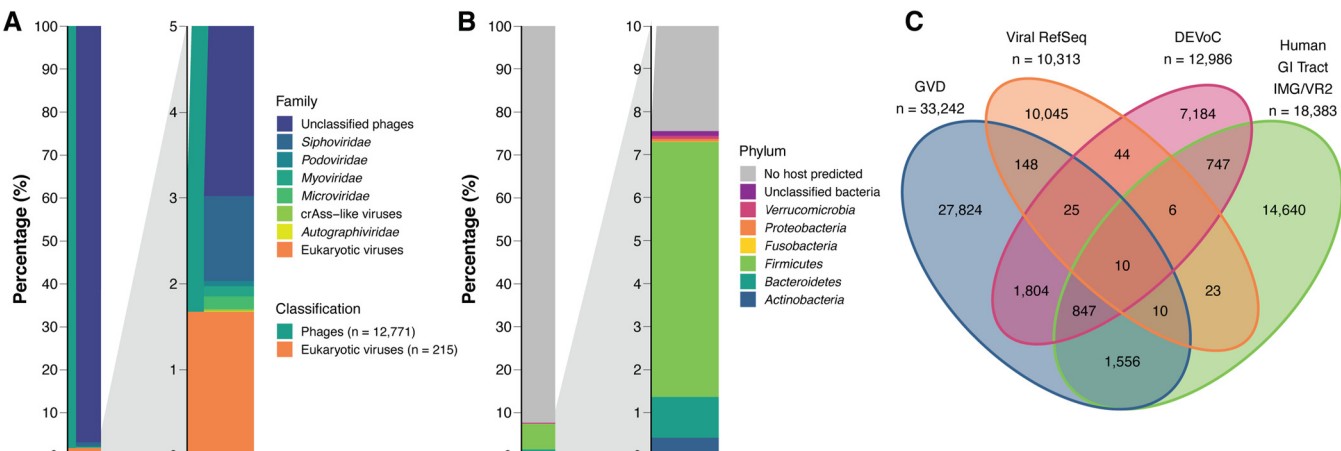

**FIG 1** DEVoC mainly consists of undescribed phages. (A) Overview of the DEVoC scaffolds ($n$ = 12,986) by type of virus and phage family. (Breakdown of the eukaryotic viruses into families is visualized in Fig. S3). (B) Overview of the DEVoC phages ($n$ = 12,771) by phylum of the predicted bacterial host. (C) Venn-diagram showing the number of clusters with members of the DEVoC, GVD, IMG/VR2, and ViralRefSeq databases at 95% identity over 80% coverage. Numbers in the Venn diagram do not sum up to the database sizes, as a viral sequence from one database may cluster with multiple partial sequences from a second database; 2,319 sequences in DEVoC, 1,018 in GVD, 544 in IMG/VR2, and 2 in ViralRefSeq were merged in this manner.

most prevalent viruses of the human gut (51, 52). Additionally, a recent study showed that human gut viromes are highly individual (29), emphasizing the importance of cataloging viromes from diverse human populations.

In this study, we characterized 254 fecal viral metagenomes from Danish children and adolescents (6 to 18 years old) and adults (aged 40 to 73 years old), to develop the Danish Enteric Virome Catalog (DEVoC). The DEVoC facilitated assessment of the diversity of the healthy Danish gut viromes, a population in which gut viromes have not been characterized before. Some phage genomes (PGs) were associated with age, while other PGs were present in human gut viromes worldwide. In particular, a previously undescribed PG, which we named LoVEphage, was prevalent in both the healthy Danish subjects and in publicly available human gut viromes. These insights, as well as the DEVoC, will further improve our understanding of the role of viruses in the human gut microbiota and thus human health.

## RESULTS

**A catalog of 12,986 nonredundant viral scaffolds derived from Danish fecal viromes encoding 190,029 proteins.** The Danish Enteric Virome Catalog (DEVoC) was constructed based on 254 Danish fecal viromes (3.86 billion raw reads). The viral scaffolds constituting the DEVoC ranged in size from 1 kb to 191 kb ($N_{50}$, 16 kb; $L_{50}$, 1,463 scaffolds), of which 1,867 viral scaffolds (14.4%) were more than 50% complete as estimated by CheckV (53). This small subset of viral scaffolds, however, dominates these Danish fecal viromes, as they represented 87.4% of the total amount of viral reads (Fig. S1A).

Phages represented the vast majority of DEVoC scaffolds ($n$ = 12,771; 98.3%; Fig. 1A) and viral reads (99.2%). The phage scaffolds were clustered using vConTACT2 to generate viral clusters (VCs) as a proxy for viral subfamilies or genera. vConTACT2 formed 1,488 VCs covering 5,222 phage scaffolds (41% of the DEVoC phage scaffolds) representing 73% of the phage reads (Fig. S1B). Merely 176 phage scaffolds (1.4%) could be taxonomically classified based on clustering with RefSeq genomes (30 VCs)—3 crAss-like genomes (1 VC), 19 *Microviridae* genomes (3 VCs), 1 *Autographiviridae* genome (1 VC), 8 *Podoviridae* (3 VCs), 16 *Myoviridae* (6 VCs), and 129 *Siphoviridae* genomes (16 VCs). Bacterial hosts were identified using CRISPR spacers for 963 phage scaffolds (7.5%). At the phylum level, *Firmicutes* ($n$ = 758) and *Bacteroidetes* ($n$ = 121) accounted for the largest fractions of hosts (Fig. 1B), while *Faecalibacterium* ($n$ = 226), *Bacteroides* ($n$ = 62), *Ruminococcus* ($n$ = 58), and *Bifidobacterium* ($n$ = 51) were the most common host genera.

A small subset of the DEVoC scaffolds represented viruses infecting eukaryotes ($n$ = 215; 1.7%). Most putative eukaryotic viral scaffolds (65.6%) belonged to the

mSystems®

*Picobirnaviridae* family (subject to interpretation, as increasing evidence suggests that viruses belonging to this family are phages [54]). The remaining putative eukaryotic viral genomes belonged to plant-infecting viral families probably originating from the diet (*Alphaflexiviridae* [0.9%], *Betaflexiviridae* [1.4%], *Bromoviridae* [1.4%], *Partitiviridae* [7.0%], *Tombusviridae* [0.5%], *Tymoviridae* [0.5%], and *Virgaviridae* [5.1%]), fungi-infecting viral families (*Chrysoviridae* [1.4%] and *Totiviridae* [2.8%]), and viral families that are known or hypothesized to infect mammals (*Anelloviridae* [0.9%], *Caliciviridae* [1.4%], *Circoviridae* [5.1%], *Genomoviridae* [2.3%], *Parvoviridae* [0.5%], *Picornaviridae* [2.3%], and *Smacoviridae* [0.5%]) (Fig. S2A).

To understand the functional potential of the viruses in the DEVoC, we predicted viral genes and annotated them using Cenote-Taker 2 (49). The 190,029 DEVoC genes ranged in size from 0.06 to 18.2 kb (median, 0.34 kb; interquartile range [IQR], 0.19 to 0.62 kb), and 91.3% were complete. About half of the DEVoC genes ($n = 102,018$; 53.7%) were functionally annotated, with the most common predicted annotations being major capsid protein, portal protein, large terminase, integrase, and minor capsid protein, all typical phage functions. The DEVoC proteins were clustered using Proteinortho (55) and formed 18,473 orthologous groups (OGs), containing up to 360 members (median, 3 members; IQR, 2 to 6 members) covering 140,581 DEVoC proteins (74%). The remaining 49,448 proteins (26%) remained singletons (regarded as OGs with one member from now on).

**The majority of the DEVoC scaffolds are previously undescribed.** We compared DEVoC scaffolds to existing viral genome databases to assess their novelty. Viral scaffolds from the NCBI RefSeq v201 database ($n = 10,313$), the human Gut Virome Database (18) (GVD; $n = 33,242$), and the human gastrointestinal tract subset of the IMG/VR2 database (56) ($n = 18,383$) were clustered with the DEVoC scaffolds at 95% identity over 80% coverage. Each of the databases contained a remarkably large set of previously undescribed viral sequence clusters (DEVoC, 67.3%; GVD, 86.3%; IMG/VR, 82.1%; ViralRefSeq, 97.4%; Fig. 1C). DEVoC shared the largest number of clusters with the GVD ($n = 2,686$ containing 4,583 DEVoC scaffolds), followed by IMG/VR2 ($n = 1,610$ containing 3,096 DEVoC scaffolds). Only 857 clusters (containing 1,960 DEVoC scaffolds) were shared among all three human gut-specific viral genome databases, and these were all phage clusters. This small overlap between the databases reflects the high interpersonal, potentially cross-regional, age-spanning variation of the human gut virome that metagenomic research has merely begun to uncover. A minor fraction of the DEVoC clusters was shared with ViralRefSeq ($n = 85$ containing 222 DEVoC scaffolds), 62 phage and 23 eukaryotic viral clusters. This limited overlap can be attributed to the underrepresentation of phages in ViralRefSeq (3,672 phage genomes versus 9,476 eukaryotic virus genomes). In total, all four databases shared 10 viral clusters, including an uncultured crAssphage, and members of the *Siphoviridae* (*Ceduavirus*, *Limdunavirus*, *Oengusvirus*, *Skunavirus*, and *Unaquatrovirus* genera) and *Myoviridae* (*Brigitvirus*, *Lagaffevirus*, *Peduovirus*, and *Toutatisvirus* genera) families (Table S1).

**Healthy Danish gut viromes are highly individual.** The remaining analyses solely included gut viromes from 91 healthy Danish subjects, including 46 children and adolescents (6 to 18 years old) and 45 adults (40 to 73 years old). Samples that we did not analyze belong to obese children and adolescents and alcoholic liver disease (ALD) patients, which are all part of a larger ongoing study. The 91 healthy Danish gut viromes were dominated by phages (relative abundance versus all viral reads; median, >99.9%; IQR, 99.8% to 100%; range, 79.4% to 100%). As multiple fragments from the same genome can hamper phage community-level analysis when they are treated as separate viruses, we restricted the analysis to phage scaffolds that represented more than 50% of a genome as determined by CheckV (53) (here referred to as phage genomes [PGs]). This allows us to limit the analysis to a maximum of one fragment for any given genome. Within the 91 healthy Danish gut viromes, 7,153 phage scaffolds (56% of the DEVoC phage scaffolds) were detected, and 1,162 of these were PGs (62.2% of DEVoC PGs). The PGs recruited a median of 90.2% of the phage reads per sample (IQR, 78.8% to 94.6%; range, 0.83% to 99.6%). The sample in which PGs

mSystems®

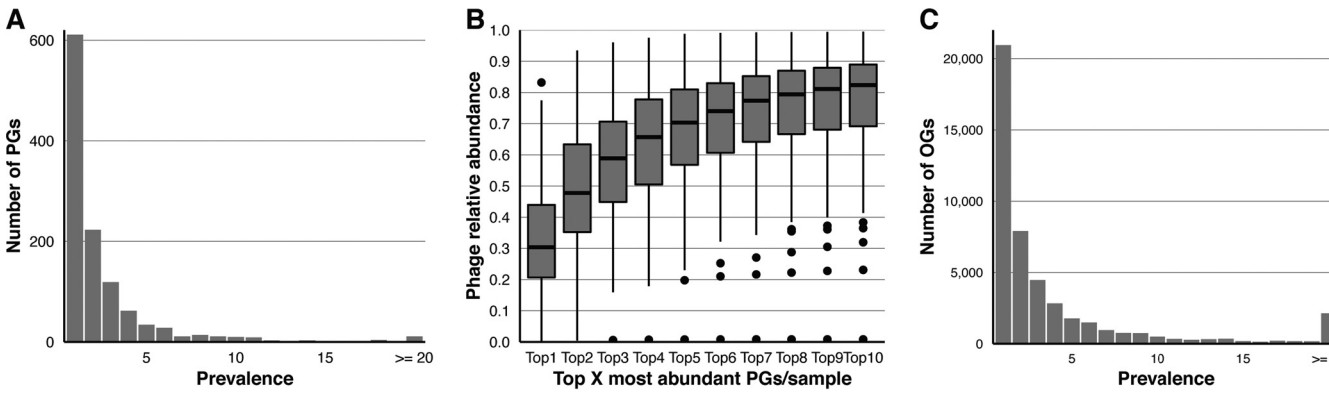

**FIG 2** Danish gut viromes are highly individual and dominated by a limited number of phages. (A) Bar plot of the prevalence of PGs (n = 1,162) in healthy Danish subjects (n = 91). PGs occurring in 20 or more subjects are grouped. (B) Boxplots of the fraction of all phage reads taken up by the most dominant PGs in different healthy Danish subjects (n = 91). (C) Bar plot of the prevalence of the viral OGs (n = 46,620) in healthy Danish subjects (n = 91). OGs occurring in 20 or more subjects are grouped.

accounted for 0.83% of phage reads was dominated by one phage scaffold with undetermined completeness (>99% of viral reads).

The most prevalent PG was a partial *Skunavirus* genome detected in 30 subjects (33% prevalence). Including this PG, only 39 PGs (3.4% of all PGs) occurred in more than 10 subjects (>12% prevalence; Table S2). This subset of 39 highly prevalent PGs will be further looked into in the next sections and included six skunaviruses, two eponaviruses, and one *Limdunavirus*, *Unaquatrovirus*, and crAss-like phage each, while the remaining 18 highly prevalent PGs remained unclassified. In contrast, more than half of the PGs were subject-specific (n = 611; 52.6%; Fig. 2A), suggesting that the healthy gut phageome is highly individual. Within each subject's phageome, the proportion of subject-specific PGs (versus all PGs; median, 18.7%; IQR, 14.0% to 24.7%; range, <0.1% to 40.0%) and their relative abundance (versus all phage reads; median, 13.5%; IQR, 7.3% to 25.3%; range, <0.1% to 83.7%) varied greatly. The most abundant PG within each subject recruited between 0.24% and 83.2% of the phage reads (median, 30.4%; IQR, 20.7% to 44.0%; Fig. 2B), while the 10 most abundant PGs represented the majority of the phage reads in most subjects (median, 82.4%; IQR, 69.2% to 89.0%; range, 0.83% to 99.5%; Fig. 2B). This suggests that the overall diversity of the phageome can be captured by the 10 most abundant PGs in most samples.

Few eukaryotic viral species were detected in the gut viromes of healthy subjects (n = 33). The majority (n = 12) were plant viruses and therefore presumably not stable members of the gut virome but, rather, transient passengers. The median observed eukaryotic viral species richness was barely 1 (IQR, 0 to 3; range, 0 to 9; Fig. S2B), and most eukaryotic viruses were present in only one or two healthy subjects (Fig. S2C), suggesting that eukaryotic viruses are highly individual.

At the protein level, all healthy subjects combined harbored 46,620 viral OGs (68.6% of DEVoC OGs). The majority of OGs were present in only one or two healthy subjects (Fig. 2C), and the number of OGs in healthy subjects ranged from 282 to 6,397 (median, 2,270; IQR, 1,584 to 2,904). A median of 7.9% of the OGs within each subject were unique to that subject (IQR, 5.6% to 10.5%; range, 0.5 to 23.8%). Notably, the most prevalent OG was recovered in almost all subjects (n = 88; 96.7%), and 51 OGs were found across more than 80% of the subjects (Table S3). The five most prevalent OGs (prevalence, >93%) were predicted to encode a recombination protein, a nuclease, a reverse transcriptase, a terminase large subunit, and a dUTPase.

**Several phage genomes and viral functions are associated with age.** We investigated if the virome composition differed between the healthy gut phageomes of the pediatric (n = 46) and the adult cohort (n = 45) cohorts. PG alpha diversity was not affected by age group (Wilcoxon-test; observed richness, P = 0.89; Shannon's diversity, P = 0.83; Fig. S3A and B), and although age group was significantly associated with PG

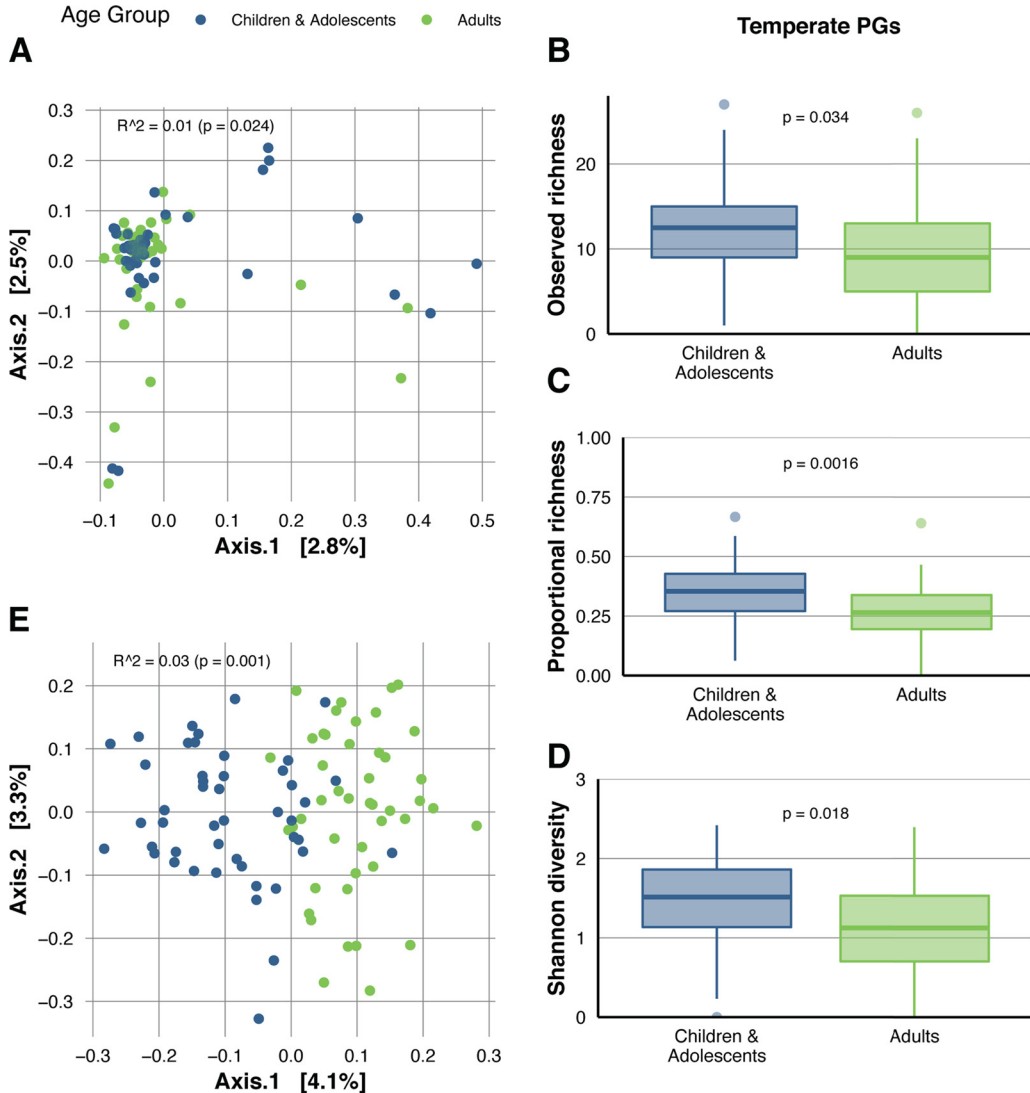

**FIG 3** Age group-associated virome patterns in healthy Danish subjects. (A) Principal-coordinate analysis on Jaccard dissimilarities between healthy Danish subjects at the PG level (PERMANOVA of age group; $R^2 = 0.01$; $P = 0.024$). Subjects are colored by age group. (B) Boxplots of the number of temperate PGs in healthy Danish children and adolescents ($n = 46$) and adults ($n = 45$) (Wilcoxon test; $P = 0.034$). (C) Boxplots of the proportional richness of temperate PGs (number of temperate PGs versus total number of PGs) in healthy Danish children and adolescents ($n = 46$) and adults ($n = 45$) (Wilcoxon test; $P = 0.0016$). (D) Boxplots of Shannon's diversity of temperate PGs in healthy Danish children and adolescents ($n = 46$) and adults ($n = 45$) (Wilcoxon test; $P = 0.018$). (E) Principal-coordinate analysis on Jaccard dissimilarities between healthy Danish subjects at the OG level (PERMANOVA of age group; $R^2 = 0.03$; $P = 0.001$). Subjects are colored by age group. All analyses are performed on 46 children/adolescents and 45 adults.

beta diversity (permutational multivariate analysis of variance [PERMANOVA]; Jaccard dissimilarity; $P = 0.024$), it only explained 1% of the variance and might hence not be biologically relevant (Fig. 3A). Low percentages of explained variability by the first two principal components indicated a large interindividual diversity in gut phageomes.

To analyze if the occurrence of individual PGs was associated with age group, we compared prevalences between the pediatric and the adult group. Among the subset of the 39 most prevalent PGs (present in more than 10 subjects; >12% prevalence), PG8 was more common in children and adolescents, while PG7 and PG22 were more prevalent in adults (Chi-square test; adjusted [adj.] $P < 0.05$; Table S2).

The genomic structures of all three age-associated PGs are visualized in Fig. S4. PG8 is predicted to encode proteins involved in the activation or suppression of the lysogenic cycle, indicating that this phage has a temperate lifestyle. Temperate phages

**TABLE 1** Orthologous groups with age-associated absence/presence profiles

| Orthologous group identifier | Size | Annotation | Function | Prevalence in the healthy subset [n (%)] | | | Chi2 test adjusted P value[a] |
|---|---|---|---|---|---|---|---|
| | | | | All (n = 91) | Pediatric cohort (n = 46)[b] | Adult cohort (n = 45)[b] | |
| OG_17005 | 180 | Hypothetical protein | Unknown | 63 (69.2) | **42 (91.3)** | 21 (46.7) | 0.0011 |
| OG_17212 | 205 | Carlavirus endopeptidase | Assembly | 63 (69.2) | **42 (91.3)** | 21 (46.7) | 0.0011 |
| OG_116 | 149 | Tail assembly chaperone protein | Assembly | 58 (63.7) | **40 (87)** | 18 (40) | 0.0008 |
| OG_17197 | 159 | Hypothetical protein | Unknown | 56 (61.5) | **42 (91.3)** | 14 (31.1) | < 0.0001 |
| OG_17367 | 149 | Major capsid/head protein | Structural | 54 (59.3) | **41 (89.1)** | 13 (28.9) | < 0.0001 |
| OG_17685 | 118 | Minor structural protein | Structural | 54 (59.3) | **39 (84.8)** | 15 (33.3) | 0.0002 |
| OG_863 | 115 | Hypothetical protein | Unknown | 49 (53.8) | **36 (78.3)** | 13 (28.9) | 0.0006 |
| OG_16990 | 86 | Hypothetical protein | Unknown | 46 (50.5) | **35 (76.1)** | 11 (24.4) | 0.0002 |
| OG_1899 | 95 | Tail completion protein | Assembly | 44 (48.4) | **35 (76.1)** | 9 (20) | < 0.0001 |
| OG_2871 | 96 | Portal protein | Packaging | 43 (47.3) | **33 (71.7)** | 10 (22.2) | 0.0006 |
| OG_3146 | 86 | Hypothetical protein | Unknown | 41 (45.1) | **32 (69.6)** | 9 (20) | 0.0005 |
| OG_3199 | 20 | Polysaccharide export protein | Other | 37 (40.7) | **31 (67.4)** | 6 (13.3) | < 0.0001 |
| OG_16319 | 71 | tRNA synthase | Translation | 35 (38.5) | **29 (63)** | 6 (13.3) | 0.0003 |
| OG_2045 | 48 | Hypothetical protein | Unknown | 34 (37.4) | **28 (60.9)** | 6 (13.3) | 0.0007 |
| OG_2076 | 6 | Putative metallopeptidase | Other | 33 (36.3) | 5 (10.9) | **28 (62.2)** | 0.0001 |
| OG_752 | 45 | Hypothetical protein | Unknown | 33 (36.3) | **29 (63)** | 4 (8.9) | < 0.0001 |
| OG_16058 | 4 | Hypothetical protein | Unknown | 32 (35.2) | **28 (60.9)** | 4 (8.9) | 0.0001 |
| OG_17591 | 68 | Hypothetical protein | Unknown | 31 (34.1) | **27 (58.7)** | 4 (8.9) | 0.0002 |
| OG_2749 | 34 | Hypothetical protein | Unknown | 31 (34.1) | **26 (56.5)** | 5 (11.1) | 0.0012 |
| OG_1811 | 3 | Plasmid recombination enzyme | Recombination | 30 (33) | **26 (56.5)** | 4 (8.9) | 0.0004 |
| OG_16535 | 37 | LytR response regulator | Other | 29 (31.9) | **25 (54.3)** | 4 (8.9) | 0.0009 |
| OG_17358 | 59 | Hypothetical protein | Unknown | 27 (29.7) | **25 (54.3)** | 2 (4.4) | 0.0001 |
| OG_17353 | 63 | Bromodomain RACK7-like subfamily | Other | 26 (28.6) | **24 (52.2)** | 2 (4.4) | 0.0001 |
| OG_18041 | 45 | Head tail connector protein | Structural | 26 (28.6) | **24 (52.2)** | 2 (4.4) | 0.0001 |
| OG_18129 | 44 | Minor structural protein | Structural | 24 (26.4) | **22 (47.8)** | 2 (4.4) | 0.0008 |
| OG_2613 | 38 | Hypothetical protein | Unknown | 22 (24.2) | **21 (45.7)** | 1 (2.2) | 0.0004 |
| OG_3028 | 35 | Hypothetical protein | Unknown | 22 (24.2) | **21 (45.7)** | 1 (2.2) | 0.0004 |
| OG_2406 | 37 | DNA binding protein | Other | 20 (22) | **20 (43.5)** | 0 (0) | 0.0002 |
| OG_2150 | 27 | Hypothetical protein | Unknown | 19 (20.9) | **19 (41.3)** | 0 (0) | 0.0004 |

[a]Bonferroni-adjusted P values of chi-squared test on prevalences.
[b]Prevalences in bold indicate the cohort with the highest prevalence.

have the potential to alter the bacterial host phenotype and shift the dynamics of the complex gut microbial network. Therefore, we identified lysogeny-associated genes (listed in Table S4) in the PGs and classified 345 temperate PGs in the healthy Danish subjects (29.7%). Each subject had a median of 11 different temperate PGs (IQR, 6.5 to 15; range, 0 to 27), representing roughly one-third of a subject's PGs (median, 31.6%, IQR, 21.5% to 39.5%; range, 0 to 66.7%) and accounting for a median of 19.3% of the PG reads (IQR, 8.8% to 42.6%; range, 0% to 95.3%). Among the temperate PGs, the alpha-diversity measures observed (absolute) richness, proportional (versus all PGs) richness, and Shannon diversity were higher in children/adolescents than in adults (Wilcoxon test; $P = 0.034$, $P = 0.0016$ and $P = 0.018$, respectively; Fig. 3B to D), while we did not observe a difference in the relative abundance of temperate PG (versus all phage reads; $P = 0.21$; Fig. S3C).

We further assessed the association between age group and viral functions represented by OGs. Similar to the previous analysis, the observed richness of viral OGs did not differ between age groups (Wilcoxon test; $P = 0.11$; Fig. S3D). However, age group explained 3% of the beta diversity between subjects (Jaccard dissimilarity; PERMANOVA; $P = 0.001$; Fig. 3E). Analysis of all OGs containing two or more members and present in more than 10 healthy subjects ($n = 3,627$) identified 29 OGs with a higher prevalence in one or both age groups (Chi-squared test; adj. $P < 0.05$; Table 1). Interestingly, only one OG (a putative metallopeptidase) was detected more often in adults, while the remaining OGs were more common in the pediatric cohort.

**Highly prevalent DEVoC phage genomes are detected worldwide.** We further examined whether the subset of 39 highly prevalent PGs defined earlier in the healthy Danish subjects (Table S2) could be recovered worldwide, across age groups and

diseases. For this purpose, we obtained 1,880 fecal viral metagenomes from NCBI SRA (denoted SRA viromes here), deriving from 1,181 subjects (see Table S5 for an overview of the included studies). The highly prevalent DEVoC PGs were widely detected in SRA viromes (Fig. 4A). The prevalence of these 39 PGs was significantly associated with the geographical region (continent of sample collection) of the SRA viromes (Kruskal-Wallis test; $P < 0.0001$; Fig. 4B). Our prevalent PGs were found more often in Europeans ($n = 164$) than in subjects from the other continents (Wilcoxon signed-rank test; versus America ($n = 170$), adj. $P < 0.0001$; versus Africa ($n = 188$), adj. $P < 0.0001$; versus Asia ($n = 20$), adj. $P = 0.038$). Moreover, they exhibited higher prevalence in Americans than Africans (Wilcoxon signed-rank test; adj. $P < 0.0001$). Age groups were also significantly associated with the prevalence of these PGs (Kruskal-Wallis test; $P < 0.0001$; Fig. 4C). Children and adolescents (3 to 17 years old; $n = 12$) had the lowest prevalence (Wilcoxon signed-rank test; versus infants [0 to 2 years old; $n = 159$], adj. $P = 0.0054$; versus adults [18 to 64 years old; $n = 231$], adj. $P < 0.0001$; versus elderly [≥65 years old; $n = 38$], adj. $P = 0.0001$), followed by infants (Wilcoxon signed-rank test; versus adults, adj. $P < 0.0001$; versus elderly, adj. $P = 0.0025$). We did not observe a significant association between healthy ($n = 472$) and all diseased ($n = 247$) subjects (Wilcoxon rank sum test; $P = 0.13$). The type of disease did, however, have an effect (Kruskal-Wallis test; $P < 0.0001$; Fig. 4D). Remarkably, malnourished Malawian infants ($n = 12$) lacked all 39 highly prevalent PGs, and consequently, prevalence was significantly lower in this group than in all other disease groups besides the HIV patients (Wilcoxon signed-rank test; versus inflammatory bowel disease [IBD, $n = 48$], adj. $P = 0.0044$; versus type 1 diabetes [T1D, $n = 29$], adj. $P = 0.0117$; versus adenoma [$n = 28$], adj. $P = 0.0010$; versus *C. difficile* infection [CDI, $n = 35$], adj. $P = 0.0035$; versus colorectal carcinoma [CRC, $n = 28$], adj. $P = 0.0056$; versus hematopoetic stem cell transplantation [HSCT, $n = 44$], adj. $P = 0.0004$). Furthermore, patients undergoing HSCT ($n = 44$) had a higher prevalence than T1D ($n = 29$; adj. $P = 0.0077$), IBD ($n = 48$; adj. $P = 0.0003$), and HIV patients ($n = 22$; adj. $P = 0.0246$).

**A crAss-like phage and a previously undescribed phage were highly prevalent in healthy Danish subjects and shared across the world.** Among the 39 most prevalent PGs in the healthy DEVoC subset, two were widely distributed in SRA viromes (Fig. 4A). A 99-kb circular crAss-like phage (PG2) was the most prevalent in SRA viromes (20.6% prevalence; Fig. 5A). CrAssphages infect *Bacteroidales* sp. and are among the most abundant and globally distributed group of viruses in the human gut (51, 52). The second most prevalent PG in SRA viromes (PG6; 14.4% prevalence; Fig. 5B), was a 71-kb circular phage without clear homology to previously described phages. Despite the lack of clear homology, this PG possessed l̲ots o̲f v̲iral (genetic) e̲lements and was therefore named LoVEphage. The prevalence of these two phages was associated with age group and geographical location (test of equal proportions between multiple groups; $P < 0.001$ for both age group and geographical location for both PG2 and PG6). None of the two PGs were detected in healthy children/adolescents from other studies ($n = 12$), although they were detected in the DEVoC healthy children/adolescents. While they occurred in, respectively, 7.5% and 5% of the infants ($n = 159$), their prevalence significantly increased to 32.5% and 20.8% in adulthood ($n = 231$; test of equal proportions; PG2, adj. $P < 0.00001$; PG6, $P = 0.00015$) and to 42.1% and 28.9% in the elderly ($n = 38$; test of equal proportions; PG2, adj. $P < 0.0001$; PG6, $P = 0.00015$). The crAss-like phage was significantly more prevalent in healthy Europeans ($n = 164$; 34.8% prevalence) and healthy Asians ($n = 20$; 50% prevalence) than in healthy Americans ($n = 170$; 21.1% prevalence; test of equal proportions; adj. $P = 0.02445$ versus Europeans; adj. $P = 0.02445$ versus Asians) while less prevalent in healthy Africans ($n = 118$; 3.4% prevalence; test of equal proportions; adj. $P < 0.001$ versus all other continents). The LoVEphage was more prevalent in healthy Europeans ($n = 164$; 20.7% prevalence) and Americans ($n = 170$; 18.8% prevalence) than in healthy Africans ($n = 118$; 2.5% prevalence; test of equal proportions; versus Europeans, adj. $P = 0.00011$; versus Americans, adj. $P = 0.00035$). Asians ($n = 20$) had a prevalence of 15% for the LoVEphage (PG6). Additionally, we found that the prevalence of the crAss-

mSystems®

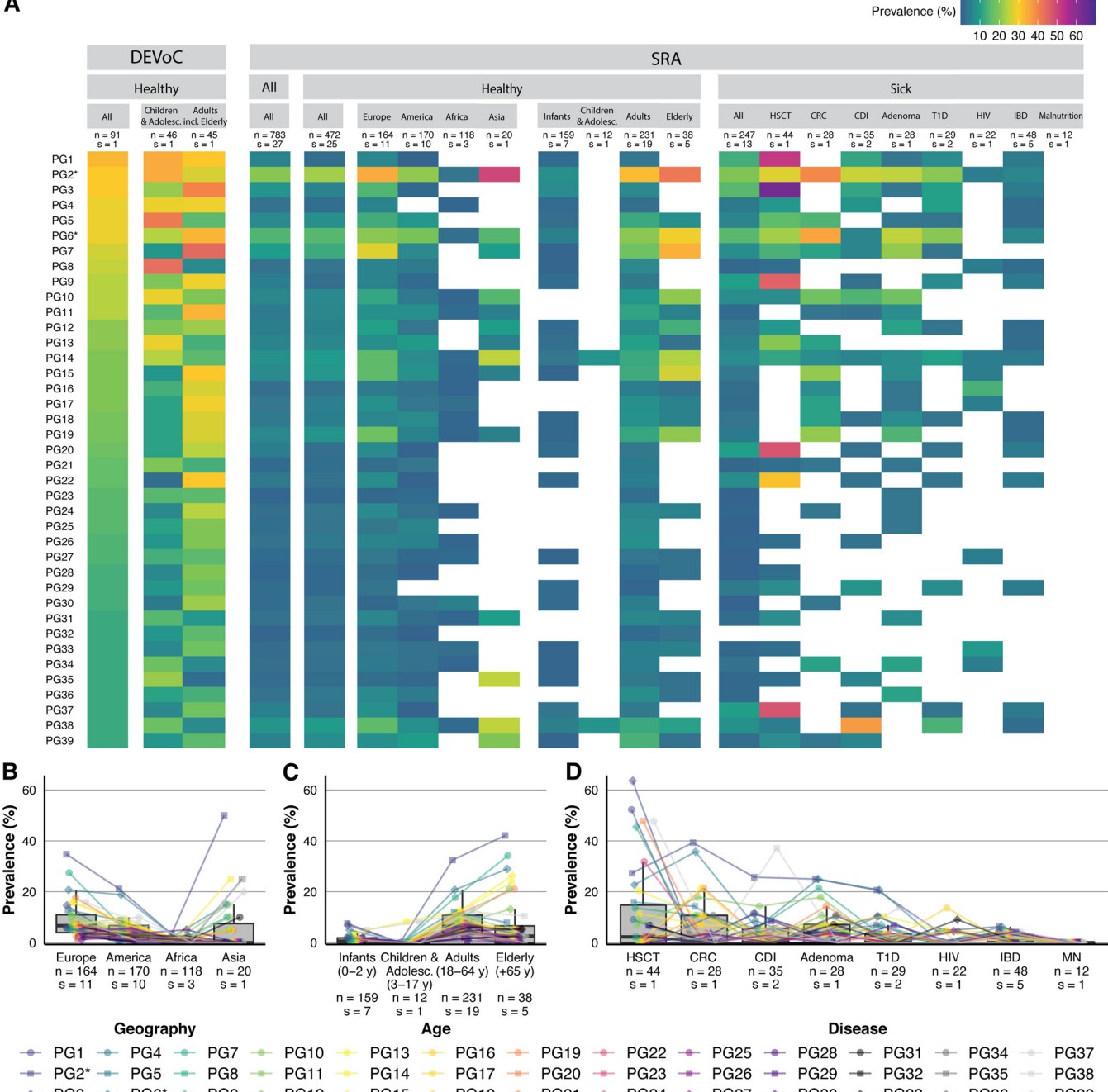

**FIG 4** Worldwide prevalence of the 39 most prevalent healthy Danish PGs. (A) Heatmap of the prevalence of the top 39 most prevalent PGs (rows) in different subsets of subjects (columns) from this study's healthy Danish population (columns 1 to 3) and from other human gut virome studies (columns 4 to 19). The first four columns represent the prevalence in healthy Danish subjects, all healthy Danish children and adolescents (6 to 18 years old), and all healthy Danish adults, including the elderly (40 to 73 years old) from the DEVoC cohort. The fourth column shows the overall prevalence in all subjects from all the other studies combined. Columns 5 to 13 represent the prevalence in the healthy subjects (column 5), separated by continent (column 6 to 9) and age group (columns 10 to 13). Columns 14 to 22 represent the prevalence in disease subjects (column 14) in different diseases (column 15 to 22). The numbers of included subjects (n) and studies (s) are indicated on top of each column. PGs not detected in a specific subset are marked by a blank square. (B) Boxplots showing the prevalence of the top 39 PGs in different continents. (C) Boxplots showing the prevalence of the top 39 PGs in different age groups. (D) Boxplots showing the prevalence of the top 39 PGs in different diseases. Prevalences are indicated by different shapes and colors by PG and connected across boxplots in panels B, C, and D, and the numbers of subjects (n) and studies (s) included in each subgroup are indicated below each boxplot. PGs with an asterisk are further discussed in Fig. 5. HSCT, hematopoietic stem cell transplantation; CRC, colorectal cancer; CDI, *Clostridium difficile* infection; T1D, type 1 diabetes; HIV, human immunodeficiency virus; IBD, inflammatory bowel disease; MN, malnutrition.

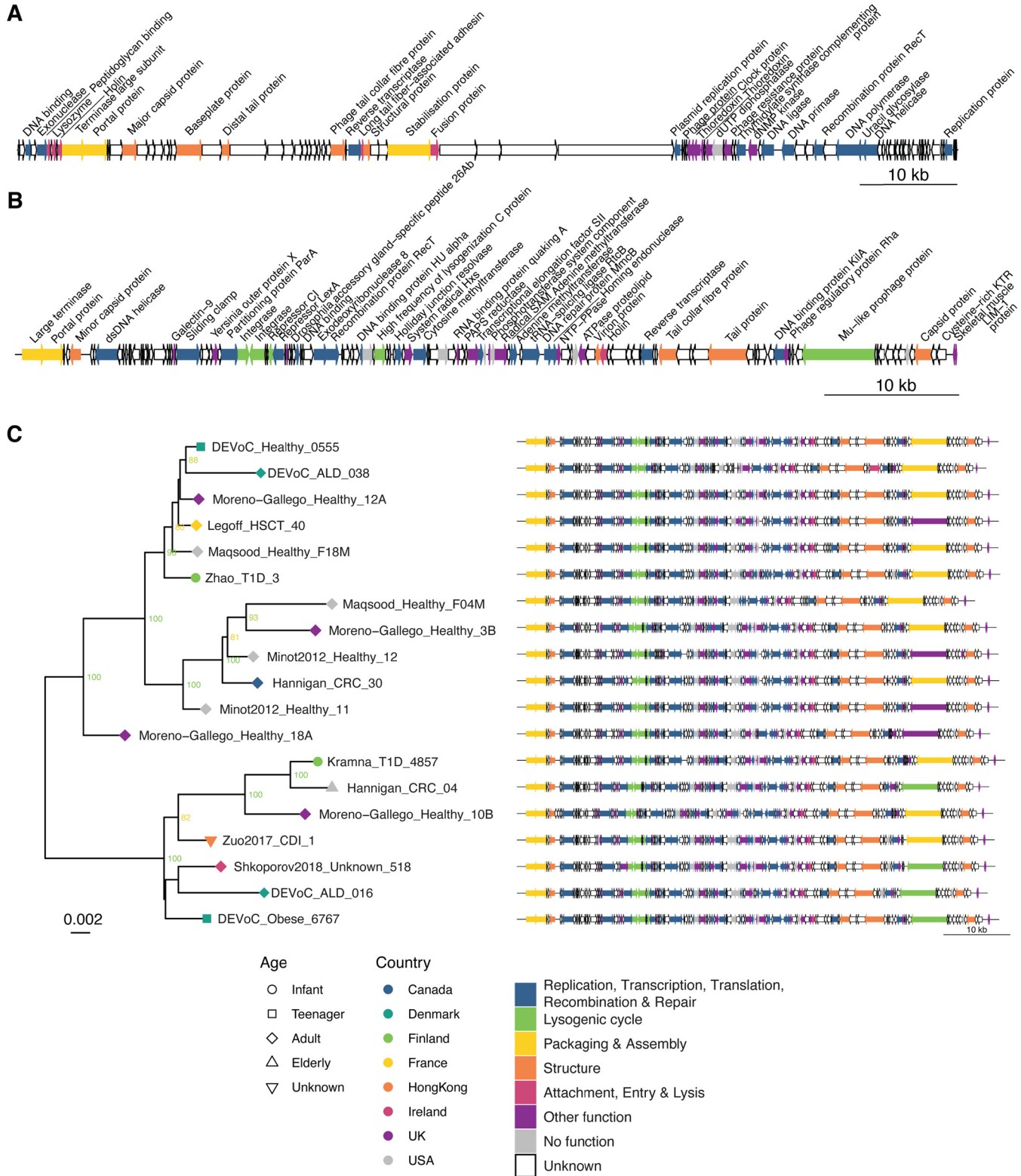

**FIG 5** Genomic structure of the identified crAss-like and LoVEphage genomes. (A) The genomic structure of a crAss-like phage (PG2). (B) The genomic structure of the LoVEphage (PG6). (C) Maximum likelihood trees of concatenated protein alignments (*n* = 61) of 19 LoVEphage-like genomes recovered from Danish subjects and SRA viromes. Only bootstrap values higher than 70 are shown. Tip symbol and color are indicative of, respectively, the age group and country of origin of the individual from which the LoVEphage-like genome was assembled. Disease states are indicated in the tip labels. Full-genome structures of the 19 LoVEphage-like genomes are visualized next to the phylogenetic tree. ALD, alcohol liver disease; CRC, colorectal cancer; T1D, type 1 diabetes; HSCT, hematopoietic stem cell transplantation. All genomes in panels A, B, and C are represented linearly for clarity, although all have a circular genome. Arrows indicate ORFs, and annotations for known ORFs are given. Unknown proteins indicate ORFs with no hit to any of the databases. ORFs without function indicate proteins with hits to hypothetical proteins.

like phage was affected by disease (test of equal proportions between multiple groups; $P = 0.004$). Remarkably, its prevalence was significantly lower in IBD patients ($n = 48$; 6.3% prevalence) than in CRC patients ($n = 28$; 39.3% prevalence; test of equal proportions; adj. $P = 0.029$), while other diseases did not affect its presence.

The circular crAss-like phage (PG2) genome of 99 kb encoded 99 proteins, of which 32 (32.3%) were functionally annotated (Fig. 5A). PG2 was classified as the *AlphacrAssvirinae* subfamily, genus *I*, one of the most prevalent gut viruses in Western subjects independent of age. Typical of a crAssphage, PG2 was subdivided into two regions with opposite gene orientation, one region encoding proteins predicted to be involved in host interaction and phage structure and the other region encoding proteins predicted to be involved in DNA replication, recombination, and nucleotide metabolism. Downstream of the tail collar fiber protein we observed a reverse transcriptase—indicative of a diversity-generating retroelement previously described in crAssphages (57). No gene was annotated as RNA polymerase; however, we suspect that one of the large unknown genes may encode a divergent RNA polymerase subunit, as large unannotated proteins in crAss-like phages often contain an amino acid motif typical for RNA polymerases (58). PG2 had no tRNA genes, otherwise commonly found in genus *I*, *II*, and *IV AlphacrAssvirinae*.

The LoVEphage (PG6) had a circular 71-kb genome encoding 130 proteins, of which 45 (34.6%) were functionally annotated. Nine tRNA genes were identified in the LoVEphage, and the orientation of the genes was more random than that of the crAss-like phage. In this genome, we also observed a tail collar fiber protein, located upstream of a reverse transcriptase, similar to what we observe in the crAss-like phage (PG2). The predicted presence of two integrase proteins, a repressor protein, and a prophage protein suggest that the LoVEphage is a temperate phage. Furthermore, the genome was highly similar to *Bacteroides dorei* strain CL03T12C01 (GenBank accession number CP011531.1) (95.6% nucleotide identity and 96% coverage), indicating that a LoVEphage-like phage has occurred as prophage in this bacterial genome. Additionally, the *Bacteroides* genus was also predicted to be the host for the LoVEphage based on matches with CRISPR spacers.

To investigate the genetic diversity of the LoVEphage (PG6), 18 additional complete LoVEphage-like genomes were reconstructed from the DEVoC (3/18) and SRA (15/18) viromes. Each complete genome (67.9 to 72.4 kb) encoded between 122 and 131 genes, of which 61 conserved proteins were selected for phylogenetic analysis based on concatenated protein alignment. Two large phylogenetic clusters can be distinguished (Fig. 5C). The largest cluster contains 12 genomes mainly obtained from healthy adults, while the smaller cluster contains 7 genomes from subjects with variable ages and disease states. However, no distinct clustering based on geography, age, or health status was observed. All 19 genomes show remarkable conservation of synteny (Fig. 5C). The largest gene in these genomes has a conserved position but has one of three annotations. Four proteins, including the protein from the reference, are annotated as "mu-like prophage protein" (indicative of temperate phages and involved in tail assembly; green), while 11 proteins are annotated as "tail tape measure protein" (involved in tail assembly; yellow) and four proteins, as "reticulocyte binding protein rhoptry" (a protein involved in the entry of the malaria parasite in red blood cells; purple). Proteins from the latter two groups show >88% amino acid identity to the proteins annotated as "mu-like prophage protein," while they show >99% pairwise amino acid identity. Therefore, we assume that "reticulocyte binding protein rhoptry" is likely a misannotation in one of the databases used.

## DISCUSSION

Human gut viruses represent a major pool of diverse and relatively underexplored microbes that, together with other gut microbiota, are believed to impact human health and disease (59). Currently, the number of studies exploring the human gut

viruses and cataloging their viral genomes and genes is expanding significantly, collectively advancing the virome field.

In this study, a human enteric virome catalog (the DEVoC), containing 12,986 viral scaffolds and encoding 190,029 genes, was generated from 254 fecal viromes from Danish children, adolescents, and adults. The majority of the DEVoC scaffolds originated from unclassified phages (Fig. 1A) without an assigned bacterial host (Fig. 1B), as described in other human gut virome databases (18). Even though the viral RefSeq version used during vConTACT2 clustering contained the most recently established phage families *Ackermannviridae*, *Herelleviridae*, *Chaseviridae*, *Dexlerviridae*, and *Demerecviridae*, none of the DEVoC scaffolds formed a viral cluster (VC) with these families. Although less stringent taxonomical classification approaches could increase the number of phage genomes with assigned taxonomy, a large fraction of phage scaffolds would remain unclassified nonetheless, hampering potential subsequent analyses at the family/genus level. Hence, further analyses were conducted at the individual scaffold level, thereby also avoiding having the results become outdated due to the constantly evolving phage taxonomy. DEVoC phages (*Caudovirales* and *Petitvirales* orders) and their bacterial hosts (*Firmicutes* and *Bacteroidetes* phyla) have all been commonly described in the human gut (18, 29, 60, 61).

Recent human gut virome studies concluded gut viromes of healthy Western adults to be highly individual (27, 29, 62). Individual-differentiating factors likely include geographical origin (63), age (18), diet (26, 62), and health status (11, 14, 15, 64, 65). Hence, our findings of large individuality of the gut viromes in healthy Danish adults is expected. The virome composition of healthy children (>3 years) and adolescents has not been studied before but is expected to show similar subject specificity since gut virome individuality has also been observed in infants (25). Due to this high virome individuality, it is not surprising that the majority of the identified viral genomes were not previously described, indicating that we are only scratching the surface of the viral diversity in the human gut microbiota worldwide (Fig. 1C). Notably, the very limited overlap of DEVoC with viral RefSeq indicates the clear underrepresentation of gut phages in the RefSeq database.

The DEVoC scaffolds encoded 190,029 genes, of which 53.7% could be annotated. However, exact estimates of functions were impeded, as multiple descriptions of the same function exist. To overcome this issue and the problem of unannotated proteins in general, proteins were clustered into OGs which were used as a proxy for function. As there is currently no database cataloguing the proteins encoded by gut viral genomes, the DEVoC encoded proteins could serve as a starting point to study the functional capacities of human gut viromes.

We further characterized the gut viromes in a subset of Danish healthy children, adolescents, and adults (*n* = 91) used to develop the DEVoC. The substantial number of previously undescribed DEVoC viral genomes is a clear indication of high individuality of human gut viromes and was further reflected by the low prevalence of most PGs (phage genomes predicted to be at least 50% complete) (Fig. 2A). It should be noted that in each healthy subject the majority of the viral reads belonged to only a few phage genomes (Fig. 2B). Despite this virome individuality, we identified 39 PGs present in more than 10 healthy subjects (>12% prevalence) with a maximum prevalence of 33% (30 subjects) (Table S2). This finding refutes the existence of a "core" virome (phages present in >50% of subjects) (66)—at least at genome level—in line with previous studies (18). OGs, on the other hand, were much more prevalent and could be detected in up to 97% of the healthy subjects (Table S3); most of these are involved in typical phage functions. However, similar to previous findings (67), the majority of the OGs remained specific to only one subject (Fig. 2C).

Gregory et al. (2020) reported an age-dependent virome diversity using publicly available data (18). They included studies produced with various wet-lab procedures and sequencing depths, as well as age groups with unequal age ranges and sample sizes (infants [<3 years], *n* = 27, versus children/adolescents [3 to 18 years], *n* = 11, versus adults [18 to 65 years], *n* = 93 versus elderly [>65 years], *n* = 20). Our study could

mSystems®

not confirm the former finding, as the PG richness and Shannon diversity did not differ across age groups (Fig. S3A and B). While our study has the advantage of consistently processed samples of different age groups (range, 6 to 73 years), we lacked data from infants, young children (<6 years old), and young adults (19 to 39 years old) to make associations with age as a continuous variable. OG richness was, similar to PG richness, not different between age groups (Fig. S8). Beta diversity at the PG and OG levels were associated with age group, although the biological importance of this effect is probably limited (Fig. 3A and E). Interestingly, at the level of individual OGs and PGs, 45 OGs and 3 PGs had different prevalences across age groups (Table 1 and Table S2). The presence of age-associated PGs may indicate that some "common" (prevalence between 20 and 50%) or even "core" (prevalence higher than 50%) phages (66) might exist in smaller, more homogeneous, populations, although core phages do not exist for the general healthy human population. Moreover, while age does not seem to affect overall diversity of the gut virome, age seems to affect the presence of certain viruses. The association of specific phages with age group might be linked to the gut microbiota with the human host, affecting human host metabolism and immune response.

We observed a clear decrease in the number and proportion of temperate PGs in our healthy adult population (Fig. 3B and C). This is accordant with the finding from L. Beller, W. Deboutte, S. Vieira-Silva, G. Falony, R. Yhossef Tito, L. Rymenans, C. Kwe Yinda, B. Vanmechelen, L. Van Espen, D. Jansen, C. Shi, M. Zeller, P. Maes, K. Faust, M. Van Ranst, J. Raes, and J. Matthijnssens (unpublished data) that demonstrates a decrease in the proportion of temperate phages across the first year of life in infants and suggests that this decrease continues during childhood into adulthood. It should, however, be noted that the identification of phage genomes with the potential to enter the lysogenic life cycle will be underestimated, as not all lysogeny-associated genes are currently known, and genes could also be encoded on the missing fragments of partial phage genomes.

Finally, we investigated the prevalence of the 39 most prevalent healthy Danish PGs in worldwide gut virome studies (Fig. 4A). Geography, age, and disease were all associated with the prevalence of the top 39 PGs (Fig. 4B to D). However, the conclusions should be interpreted cautiously, as subsets consisted of heterogeneous sample sizes. Some patient subsets showed a remarkably high prevalence (e.g., CRC or HSCT patients) or complete absence (malnourished Malawian infants) of the top 39 PGs. These striking differences are possibly confounded, as they often consist of a limited number of samples from only a single study, which can cause a severe bias regarding sample preparation, sequencing depth, or study setup. The top 39 PGs were, nonetheless, most commonly found in other European adults, which could be expected given the demographic similarity to our cohorts. The top 39 PGs were less commonly observed in infants, which are known to have a more distinct gut virome composition, and this age group was not included in the development of the DEVoC. Prevalences from healthy children and adolescents should be interpreted cautiously, as this SRA subset contained very few subjects due to the limited availability of these samples. For the same reason, the age-specific PGs could not be confirmed within the SRA viromes.

The group of crAss-like phages and their high prevalence and abundance across human gut viromes have been described extensively (51, 68–70). Although not as widespread as the crAss-like phages, the newly discovered LoVEphage seems to be rather common as well, with a prevalence of 28.6% in the healthy Danish subjects and 14.4% in the SRA viromes (Fig. 4A). However, the prevalence of crAss-like phages and the LoVEphages across SRA viromes is probably an underestimate due to the stringent criteria used and the low sequencing depths of some samples. Despite not having clear homology to previously described phages, numerous typical phage genes were identified in the LoVEphage (Fig. 5B). Phylogenetic analysis of 19 LoVEphage genomes did not reveal any clustering based on age, geography, or disease status (Fig. 5C), in contrast to the crAss-like phages, which seem to have some level of local geographic

clustering (71). However, such patterns may become apparent when more LoVEphage-like genomes are included/investigated. Future studies should experimentally determine the host range and morphology of the LoVEphage, as well as their broad genetic diversity in the general population, to uncover the potential associations of variants with disease. It could be worthwhile to also investigate whether this phage is specific to humans or is also found in nonhuman primates and other mammals as is the case for crAss-like phages (71, 72).

In conclusion, the human gut virome catalog DEVoC and its encoded genes generated from Danish children, adolescents, and adults assisted in the characterization of the healthy gut virome and will prove very helpful in investigating the role of the gut virome in human health and disease in the future. Furthermore, by investigating the presence of the top healthy Danish PGs in other human gut virome studies, we identified a previously undescribed phage, called LoVEphage, with a high worldwide prevalence.

## MATERIALS AND METHODS

**Subject recruitment and sample collection.** The two Danish cohorts involved in this study were included as part of the MicrobLiver project. The pediatric cohort included 50 children and adolescents (6 to 18 years old) with a BMI above the 90th percentile, together with 50 age- and sex-matched healthy controls (73). The obese pediatric subjects were enrolled in an obesity treatment, and samples were included at baseline and the 1-year follow-up. The adult cohort (34 to 76 years old) included 52 patients with alcohol-related liver disease (ALD) and 52 sex, BMI, and age-matched healthy controls. They represent a selection of participants from a study aimed to develop noninvasive markers of early-stage alcohol-related liver disease. In total, 254 fecal samples were collected from 204 subjects.

Fecal samples were collected at home and kept at −20°C for 0 to 4 days, after which they were brought to the clinic (frozen) and stored at −80°C. For the adult cohort, samples were aliquoted at −120°C with the CryoXtract CXT350 device (CryoXtract Instruments) (74). Fecal samples from the pediatric cohort were aliquoted on ice, as the fecal sample sizes were much smaller. All fecal samples were kept at −80°C until use.

**Sample preparation and sequencing.** All 254 fecal samples were prepared for high-throughput virome sequencing using the NetoVIR protocol (75). In short, each fecal aliquot was homogenized in phosphate-buffered saline (PBS) (30 mass/volume percentage), centrifuged, filtered (0.8 $\mu$m), and subjected to nuclease treatment to enrich for viral-like particles. Next, the QIAamp viral RNA minikit (Qiagen) without carrier RNA was used to extract both RNA and DNA. The extracts were reverse transcribed and randomly amplified (17 cycles) using a modified WTA2 kit (Sigma-Aldrich). Sequencing libraries were prepared with the Nextera XT DNA library preparation kit (Illumina) and sequenced on the NextSeq 500 high-throughput Illumina platform (Nucleomics Core facility, KU Leuven, Belgium). Per sample, a median of 12.1 million (IQR, 6.9 million to 19.4 million) paired-end reads (2 × 150 bp) were generated.

**Development of the Danish Enteric Virome Catalog.** Raw reads were processed as described by L. Beller et al. (submitted for publication). In short, reads were quality controlled using Trimmomatic (v0.36) (76), after which reads mapping to the "contaminome" and human genome were removed using Bowtie2 (v2.3.4.1) in "very-sensitive" mode (77). Quality-filtered reads were *de novo* assembled, and all scaffolds longer than 1 kb were clustered at 95% identity over 80% coverage to remove redundancy in line with Roux et al. (78). Instead of calculating abundances by mapping the quality-filtered reads to the complete set of nonredundant scaffolds, reads were only mapped against to the representatives of the clusters containing a scaffold from that sample to avoid false-positive detection of closely related sequences. A scaffold was assumed to be present if 70% of its length was covered by reads. Scaffolds representing less than 0.00001% of the total amount of mapped reads across all samples were removed to reduce background noise. Viral scaffolds were selected to construct the Danish Enteric Virome Catalog (DEVoC). These viral scaffolds were identified by using a combination of homology to known viruses at the protein and/or nucleotide level, genome structure (kmer usage and gene content), the presence of virus-specific genes, and VirSorter category (35). The completeness of viral genomes was assessed with CheckV (v0.6.0) (53), and viral scaffolds were annotated using Cenote-Taker 2 (v2.0.1; parameters –prune_prophage False –enforce_start_codon False –hsuite_tool hhsearch) (49).

**Taxonomic classification of viral scaffolds.** Eukaryotic viruses were classified based on the lowest common ancestor determined using ktClassifyBLAST (v2.7.1) (79) on DIAMOND protein hits (v0.9.10.111, sensitive mode) (80) and BLASTn nucleotide hits (v2.7.1; E value, 1e-10) (81) (nonredundant [nr] and nucleotide [nt] databases downloaded from NCBI on 3 May 2019). As taxonomic classifications are unavailable for most phage scaffolds, vConTACT2 (v0.9.19) was used to create viral clusters (VCs) based on gene-sharing networks that represent genus/subfamily-level taxonomy (47). If phage scaffolds clustered with a RefSeq phage genome (v201), the taxonomy of the RefSeq phage genome(s) was assigned to the other members of the VC up to genus level.

**Phage host prediction.** CRISPR spacers were predicted using MinCED (v0.4.2) on the bacterial contigs assembled from shotgun metagenomic sequencing data of the same 254 fecal samples used to generate the DEVoC (unpublished data) (82). The predicted CRISPR spacers were submitted to a BLAST search against the phage subset of the DEVoC (-evalue, 1e-10, task, "blastn-short") (81). Phages required

at least two spacer matches with a maximum of one mismatch for reliable host assignment as the lowest common ancestor of the bacterial matches. Bacterial contigs were mapped against ProGenomes2 (83), and the lowest common ancestor was determined using ktClassifyBLAST (79).

**Identification and annotation of DEVoC genes.** Cenote-Taker 2 (v2.0.1) was used to predict and annotate open reading frames (ORFs) on the viral genomes of the DEVoC (49). Cenote-Taker 2 predicts ORFs using a combination of PHANOTATE (84) and Prodigal (85) in metagenomic mode and annotates the predicted ORFs using HMMER (86), RPSBLAST, (87) and HHSEARCH (88) searches against custom viral HMM, CDD, Pfam, and PDB databases. Next, amino acid sequences of the predicted ORFs were clustered into orthologous groups (OGs) using Proteinortho (v6.0.18) (55) with default settings and DIAMOND v0.9.32 (80). The annotation(s) given to at least 10% of the protein members of a specific OG was assigned to that OG of interest (manually, to overcome differences in spelling, capitalization, and abbreviations, as well as synonyms of the same protein, due to the use of different databases). This 10% threshold was introduced to avoid spurious annotations of OGs.

**Gene prevalence.** Due to the short nature of some DEVoC genes (the lower cutoff length for ORF identification was 20 amino acids) compared to the read length, we decided against a mapping approach to determine gene presence per sample. This is because short genes would be underrepresented, as a substantial fraction of the reads would only partially overlap the gene (and therefore not be assigned) compared to larger genes. Instead, we opted to count a viral gene as present when the corresponding viral genome was present (see L. Beller et al. [submitted for publication] for genome abundances). The presence of orthologous groups (OGs) was determined by grouping the prevalence information of all genes within the specific OG.

**Comparison to existing databases.** The DEVoC and its encoded genes were compared against existing (human gut) viral genome databases, including the human Gut Virome Database (GVD, version 2020/07/23) (18), IMG/VR2 (version July 2019) (56), and viral RefSeq (v201, version 10/07/2020). To determine overlap between the different genome databases (GVD, IMG/VR2, and viral RefSeq) and the DEVoC, they were clustered with ClusterGenomes (89) at 95% identity over 80% coverage (using nucmer v3.23) (90). Only IMG/VR2 sequences originating from human digestive tract samples ($n = 78,016$) were selected for comparison, and they were clustered in advance to remove redundancy (resulting in $n = 18,383$). Likewise, also viral RefSeq sequences larger than 1 kb ($n = 12,681$) were clustered in advance (resulting in $n = 10,313$). The GVD is already nonredundant (95% identity over 70% or 100% coverage depending on the type of virus) and consists of 33,242 viral genomes.

**Prevalence of viral genomes across subjects worldwide.** The prevalence of the genomes identified in the DEVoC in other subjects was assessed by mapping publicly available SRA data sets from 26 previously published human gut viral metagenomic studies (24–34, 65, 66, 91–103) and one unpublished study from our lab to the DEVoC using the Burrows-Wheeler Aligner (BWA) (v2.0pre2) (104). Reads were trimmed before mapping using Trimmomatic (v0.63; removing WTA2 and Nextera primers with the parameters 30:10:1:true and the following quality trimming parameters: HEADCROP:19 LEADING:15 TRAILING:15 SLIDINGWINDOW:4:20 MINLEN:50) (76).

An overview of all included studies is available in Table S5. Studies were selected based on a PubMed search in December 2019 searching for "human gut/fecal/enteric viromes." Studies using targeted sequencing or not using viral enrichment were excluded, as well as studies for which raw sequencing reads and/or metadata (subject ID, age group, health status, and geographic region of inclusion) were unavailable. The metadata were curated by screening the original article's subject recruitment section, supplementary tables, and/or the information association with the BioSample/SRA entry. As the gut virome is relatively stable over time (29), multiple samples from the same subject were pooled, except for patients undergoing fecal microbiota transplantation (FMT), for which only the baseline sample (before FMT) was included, if available (33, 34, 92). Two studies sequenced pools of multiple subjects, and for further analysis, these pools are regarded as one subject (91, 103). In total, 1,880 samples from 1,181 subjects (of which 490 were sequenced in 92 pools) were assessed. The subjects ranged in age from 0 (24, 25, 65, 103) to 99 (34) years old and originated from different geographical locations (13 countries across 4 continents). Besides healthy subjects, subjects suffering from inflammatory bowel disease (IBD) (30–32, 92, 101), *C. difficile* infection (CDI) (33, 34), diarrhea (91), malnutrition (105), HIV (100), type 1 diabetes (T1D) (65, 94), and colorectal cancer (CRC) (93) and subjects undergoing hematopoietic stem cell transplantation (HSCT) (95) were also included. A viral sequence was considered present in a subject if it was covered for more than 70% of its length by reads from the subject.

**CrAss-like phage genome.** To determine to which genus/subfamily the proposed prevalent crAss-like phage genome belongs (PG2), its genome is clustered with the 249 genomes from the crAss-like phage data set of Guerin et al. (70) using ClusterGenomes (89) at 95% identity over 80% coverage.

**LoVEphage genome.** To investigate the genetic diversity of the LoVEphage, we attempted to retrieve (near-)complete genomes from samples in which the LoVEphage was present. For the Danish samples, scaffolds longer than 50 kb clustering together with the LoVEphage (see above) were selected. All SRAs of subjects in which the LoVEphage was covered by reads for at least 70% of its length were quality-trimmed using Trimmomatic (76) (same settings as before) and assembled using metaSPAdes (v3.11.1; parameters -k 21,33,55,77) (106). A BLASTn search of the *de novo* assembled contigs was performed against the reference LoVEphage (E value, 1e-10) (81). All contigs larger than 50 kb, which covered the reference genome for at least 70% with a similarity of >70%, were selected. Incomplete genomes were completed using additional smaller scaffolds and/or individual quality-filtered reads (after mapping to the reference LoVEphage using BWA [104]), resulting in 18 additional complete LoVEphage genomes. Cenote-Taker 2 (v2.0.1) was used to predict and annotate ORFs on the complete LoVEphage-like genomes (same settings as before) (49). All 61 proteins that showed more than 70%

identity over 70% coverage and were present in all 19 LoVEphage-like genomes were aligned individually using MAFFT (v7.464; with automatic alignment strategy selection) (107). The individual protein alignments were concatenated and trimmed using trimAl (v1.4; parameter, -gappyout) (108). Maximum-likelihood trees were generated using RAxML (v8.2.12; parameters, -f a with 1,000 bootstraps and automatic amino acid substitution model selection) (109).

**Ecological analyses, statistical analyses, and visualization.** All ecological and statistical analyses, as well as visualizations, were done in R (http://www.R-project.org; v3.6.0). Viral reads were subsampled to a depth of 176,256 viral reads/sample, removing 21 samples with fewer viral reads, to allow unbiased characterization of the gut virome across the samples, as virome sequencing depth is equal. Random subsampling was done using the "rarefy_even_depth" function of the phyloseq package (v1.28.0) (110). Phageome analyses were conducted on phage relative abundances, while analyses of the eukaryotic viruses and at the protein level were performed on absence/presence profiles. Alpha-diversity indices (observed richness and Shannon's diversity) were calculated using the vegan package (v2.6-7) (111). Beta diversity was analyzed using the phyloseq package (v1.28.0) (110). Principal-coordinate analysis (PCoA) was used to visualize Jaccard distance. PERMANOVA was calculated using the "adonis" function from the vegan package. Medians of two groups were compared using the Wilcoxon test. Proportions of two groups were compared using the chi-squared test corrected for multiple testing using the Bonferroni method. Prevalences of the selected phage genomes across multiple sample subsets were compared using the Kruskal-Wallis test, after which *post hoc* Wilcoxon signed-rank tests (paired) were performed on each pair of groups corrected for multiple testing using the Holm method. Multiple proportions were compared using the test of equal proportions (prop.test in R), followed by *post hoc* tests of equal proportions on each pair of groups corrected for multiple testing using the Holm method. The genomic structure of individual phage genomes was visualized using the GenoPlotR package (v0.8.9) (112), the Venn diagrams using the VennDiagram package (v1.6.20) (113), and the phylogenetic trees using the ggtree package (v1.16.6) (114). Other figures were generated using the ggplot2 package (v3.3.2) (115).

**Data availability.** The virome sequencing reads supporting the conclusions of this article are in the Sequence Read Archive (SRA) with accession numbers PRJNA723467 (pediatric cohort) and PRJNA722819 (adult cohort). The DEVoC and its encoded genes, annotations, and normalized counts are available at https://doi.org/10.5281/zenodo.5173012. The LoVEphage genomes assembled from the above-mentioned BioProjects are available in GenBank under accession no. MW660583, MZ919976, MZ919981, and MZ919987. The scripts used to perform the analysis and make figures starting from the abundance table are available at https://github.com/Matthijnssenslab/ViromeCatalogue.

## SUPPLEMENTAL MATERIAL

Supplemental material is available online only.
**FIG S1**, EPS file, 1.7 MB.
**FIG S2**, PDF file, 0.4 MB.
**FIG S3**, EPS file, 1.5 MB.
**FIG S4**, PDF file, 0.2 MB.
**TABLE S1**, XLSX file, 0.01 MB.
**TABLE S2**, XLSX file, 0.01 MB.
**TABLE S3**, XLSX file, 0.01 MB.
**TABLE S4**, XLSX file, 0.01 MB.
**TABLE S5**, XLSX file, 0.01 MB.

## ACKNOWLEDGMENTS

This research was supported by the Novo Nordisk Foundation Center for Basic Metabolic Research, Faculty of Health and Medical Sciences, University of Copenhagen, Denmark (grant number NNF18CC0034900), the Challenge Grant "MicrobLiver" (grant number NNF15OC0016692), and grant number NNF15OC0016544 from the Novo Nordisk Foundation; the Innovation Fund Denmark (TARGET: grant number 0603-00484B), the Region Zealand Health Scientific Research Foundation; and the European Union's Horizon 2020 research and innovation program (GALAXY: grant number 668031); the "Fonds Wetenschappelijk Onderzoek" (FWO, Research Foundation Flanders) (Lore Van Espen: 1S25720N, Leen Beller: 1S61618N).

The computational resources were provided by the Flemish Supercomputer Center (VSC) and funded by FWO and the Flemish Government Department of Economy, Science, and Innovation.

The study was conceptualized by E.G.B., L.V.E., A.K., P.B., T.H., M.A., and J.M. C.F.-B., C.E.F., S.J., M. Kjærgaard, M.T. H.B.J., T.N., J.-C.H., and A.K. handled the collection and management of fecal samples. L.V.E. and E.G.B. managed the project. E.G.B. and L.C.

carried out the viral DNA/RNA extraction, amplifications, and library preparation. L.V.E. performed the bioinformatic processing of the reads, generated the catalog, and performed the statistical analysis in close collaboration with E.G.B., L.B., W.D., M.A., and J.M. A.F. and M. Kuhn predicted CRISPR spacers in bacterial metagenomes. L.V.E. performed SRA screening with the assistance of D.S. and L.D.C., L.V.E., E.G.B., M.A., and J.M. drafted the manuscript. All authors critically revised and approved the final version for publication.

We declare that we have no competing interests. The funders had no role in study design, data collection and interpretation, or the decision to submit the work for publication.

The study was approved by the Ethical Committees for the Region of Southern Denmark with reference numbers S-20120071, S-20160021, and S-20170087 (adult cohort) and by the Ethical Committees for Region Zeeland with reference number REG-043-2013 (pediatric cohort). All participants or their legal guardians gave consent to participate in this study.

We would like to thank the participants in The Danish Childhood Obesity Data and Biobank and the GALAXY study.

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
