## [Reviewer comments · mSystems]

A previously undescribed highly prevalent phage identified in a Danish enteric virome catalog

Lore Van Espen, Emilie Bak, Leen Beller, Lila Close, Ward Deboutte, Helene Juel, Trine Nielsen, Deniz Sinar, Lander De Coninck, Christine Frithioff-Bøjsøe, Cilius Fonvig, Suganya Jacobsen, Maria Kjærgaard, Maja Thiele, Anthony Fullam, Michael Kuhn, Jens-Christian Holm, Peer Bork, Aleksander Krag, Torben Hansen, Mani Arumugam, and Jelle Matthijnssens

Corresponding Author(s): Jelle Matthijnssens, KU Leuven

Review Timeline:

Submission Date:	March 29, 2021
Editorial Decision:	May 19, 2021
Revision Received:	August 12, 2021
Accepted:	September 2, 2021

Editor: Chaysavanh Manichanh

Reviewer(s): Disclosure of reviewer identity is with reference to reviewer comments included in decision letter(s). The following individuals involved in review of your submission have agreed to reveal their identity: Adrian Paskey (Reviewer #1); Ghjuvan Grimaud (Reviewer #3)

Transaction Report:

DOI: <https://doi.org/10.1128/mSystems.00382-21>

May 19, 2021

Prof. Jelle Matthijnssens
KU Leuven
Department of Microbiology and Immunology, Rega Institute, Laboratory of Viral Metagenomics
Leuven 3000
Belgium

Re: mSystems00382-21 (A previously undescribed highly prevalent phage identified in a Danish enteric virome catalog)

Dear Prof. Jelle Matthijnssens:

Thank you for submitting your manuscript to mSystems. We have completed our review and I am pleased to inform you that, in principle, we expect to accept it for publication in mSystems. However, acceptance will not be final until you have adequately addressed the reviewer comments.

Thank you for the privilege of reviewing your work. Below you will find instructions from the mSystemseitorial office and comments generated during the review.

Preparing Revision Guidelines

For complete guidelines on revision requirements, please see the Instructions to Authors at <https://msystems.asm.org/sites/default/files/additional-assets/mSys-ITA.pdf>. **Submissions of a paper that does not conform to mSystems guidelines will delay acceptance of your manuscript.**

Sincerely,

Chaysavanh Manichanh

Editor, mSystems

Journals Department
Reviewer comments:

Reviewer #1 (Comments for the Author):

See attached. Nice work!

Reviewer #2 (Comments for the Author):

For the article "A previously undescribed highly prevalent phage identified in a Danish enteric virome catalog" submitted to review in mSystems, in general, I have no objections to the methodology proposed by the authors. It seems very accurate and up-to-date to me. However, there are some things that I would like to see modified, clarified or improved in the text.

For example, the term "viral individuality" or "degree of viral individuality", I do not think is the best to refer to the specificity of the virobiota of each individual, if that is the case. Certainly, there is a high degree of viral specificity in the gut microbiota, but the term "viral individuality" sounds like something else.

Did the 91 Danish subjects have the same diet or at least a similar diet for a particular period of time? This could have an impact on the determination of the intestinal virome diversity of each one of them.

It is not clear to me why the authors do mention that 39 of the most frequent phage genomes in the samples are present in more than 10 subjects (> 12%). That is, the authors worked with 91 healthy individuals (46 children/adolescents + 45 adults). Why not define how many of these individuals/subjects had the 39 phage genomes? Were these 39 PGs present in all the 81 remaining subjects?

I understand that statistically, it is significant that PG8 was more abundant in children/adolescents, while PG7 and PG22 were more abundant in adults. However, PGs are still present (at least

detected in 1 subject) albeit in low proportions in its counterpart. This could be a problem due to the n of samples that authors use.

Do you have how to discuss why this difference in abundance of phages (or phage genomes) could occur in children/adolescents vs adults?

What relevance could have the fact of found differences in the abundance of PGs between children/adolescents vs adults, and how does this correlate with the title of this article?

Why in Figure 4 you separate the group of adults (45 subjects) into adults (34 subjects) and the elderly (11 subjects)? Just to compare with SRA viromes? Why not follow the same nomenclature that they used throughout the study? This separation, although it serves to compare the SRAs, would also affect the heat map of the 39 PGs.

Does the order (ascending or descending) of the graphical representation of the 19 genomes of LoVE phage (Figure 5D) follow the same order of appearing (ascending or descending, respectively) of these genomes in the phylogenetic tree in Figure 5C?

If so, why not merge both images into one?

Since authors found a high incidence of phage sequences belonging to Firmicutes and Bacteroidetes as gut hosts, I would suggest they incorporate this article to discuss their results: Koliada, A., Moseiko, V., Romanenko, M. et al. Sex differences in the phylum - level human gut microbiota composition. BMC Microbiol 21, 131 (2021). <https://doi.org/10.1186/s12866-021-02198-y>

In general, I found a very descriptive article. I would like to see a juicier discussion.

The initials of the authors should be checked in the "Author contributions" section.

Reviewer #3 (Comments for the Author):

This paper describes a new virome database named DEVOC, used here to characterize the healthy gut virome of 91 children, adolescent and adults. This effort is currently highly needed in the community, and this paper is making a very good contribution. I took genuine pleasure in reading it, although some modifications should be made.

Major comments

1) My main concern is about the availability of the data and the bioinformatic pipelines used. As I had not access to it, I have no possibilities to verify the results and the pipelines on my own. I would have been pleased with a preprint submitted to biorxiv for example, as well as access to at least the bioinformatic pipeline. I strongly recommend the authors to share the bioinformatic pipeline used in a git repository before publication (an after, of course).

2) My second concern is about the abstract. I think it could be modified to present this very nice study a little bit more clearly; it can be improved.

3) Finally, it would have been nice to have 16S amplicon sequencing data or shotgun metagenomic data associated with the study (i.e., the microbiota analysis associated). I am not asking for it, but it would certainly have added a plus-value to the paper.

Minor comments

L36-37: "The DEVoC was used to characterize 91 healthy DEVoC gut viromes": unclear. Is it 91 healthy gut viromes taken from the samples used for the database itself? Please clarify this

sentence. Also, how did you determine that the individuals are healthy?

L45-46: Why did you focus on the LOVEphage? How did you select the public datasets? How is it related to your DEVOC database? Please explain a little more here and clarify. As present, it seems to be explained/mentioned in the "importance" section of the abstract, but I think it should be explained/clarify here, otherwise it is not clear why you focus on these phages.

L47: I am not sure that "we have contributed" is the good verb to use here? Maybe "we have added".

L60-67: some references are in square brackets [] while others are in parenthesis (). Please correct it. Furthermore, there seem to be a problem in the numbering of the references, at least at the beginning of the introduction (it starts with 1, goes to 5, go back to 4 etc.)

L99: please replace "In particular, a previously undescribed PG, we named LoVEphage" by "In particular, a previously undescribed PG that we named LoVEphage"

L109-110: How did you address the incompleteness issue?

L115-116: 1.4% is very low, it would be nice to have few words about why it is so low and how to deal with it.

L 125: "plant infecting" it is probably due to the alimantation, maybe your interpretation regarding their provenance should be added.

L137: typo: add a space after the comma.

L140: "(40)using": add space

L170: precise what "ALD" is.

L239: you could have described these 39 highly prevalent PGs in the previous paragraphs, the transition here is a little bit rough

L287: please fully name the disease before to use "IBD" and "CRC"

L313: "18 additional complete LoVEphage-like genomes were reconstructed from the DEVoC and SRA viromes." How many from DEVoC, how many from SRA?

L 363: "genome level - similar to previous studies (4)." I would rather say "in line with"

L 367: "Gregory et al." please add the year in the citation.

L379: please precise the notion of core phages, it's not clear to me here.

L423: "percentile, together with 50 age- and sex-matched healthy controls²⁰". Citation should be (20) here, please homogenize.

L444: "Raw reads were processed as described by L. Beller et al. (submitted for publication)." It is not enough. I strongly suggest you share the pipelines used for this analysis.

Also, some more rational for the thresholds used would help.

L570-574: Same comment as for major concern 1)

L 896: a parenthesis is missing.

Reviewer #4 (Comments for the Author):

In the present manuscript, Espen and Bak et.al. developed a comprehensive database such as Danish Enteric Virome Catalog (DEVoC) by utilizing the human gut virome studies from 204 Danish subjects. They were able to identify the 12,986 non-redundant viral genome sequences encoding 190,029 viral genes. They performed the comparable evaluation of the viral genomes using other virome databases to reveal the unique virome composition in Danish population. Also they showed the utilization of the DEVoC by performing the characterization of healthy gut viromes from Danish people with varying ages, where they were able to dissect the phages prevalent in pediatric versus adult healthy subjects. Further, They also investigated the overall prevalent of phage genomes (PGs) by utilizing 1,880 gut virome samples of 27 studies from across the world. Finally, they observed the high prevalence of crAss-like phages affecting the *Bacteroides dorei* bacterial species.

Certainly, the work conducted has importance keeping in the view of current growing statistics of gut virome studies and to the extent they usually went underanalyzed to reveal the role of viruses and phages in the disease and also discovering novel phages. However, I do have some major suggestions regarding the manuscript in its present form to improve it to greater extent before it gets published.

Please find the section wise suggestions enlisted below

Manuscript related comments:

1. Abstract: I strongly urge the authors to restructure the abstract as its not comprehending the need and importance of the work done.

I) e.g. ln32 in the abstract seems incomplete to convey its meaning

II) ln35 we found that

III) ln47 please remove have and change sentence to "we contributed numerous previously.." and please remove all have in the manuscript. Please consider the English writing and Grammar errors throughout the manuscript.

2. Introduction is not comprehending the role of the Human gut Microbiome in Disease. Authors need to add some background history to increase the importance of their conducted work.

3. Introduction: Some key references are missing to emphasize the importance or rationale of the sentence, while some sentences don't have citations at all. See:

I) ln62 only one reference, while there are many key references shedding light on the human virome studies in the disease mechanism, here are some references/PMIDs: 32718358, 32200373, 26489866. A recent one on Inflammatory Bowel disease (IBD) is here: Whole-virome analysis sheds light on viral dark matter in inflammatory bowel disease. Cell Host Microbe. 2019 and many more as cancer immunotherapy. Authors could anchor the great importance of their work in the introduction by putting forward a well-described background and role of virome sequencing in the health; be it gut, skin Microbiome, and then mention the need of virome in Danish gut cohorts.

II) Ln. 63, 69, 73 and 74, Ln.81 references are missing,

4. What do authors meant by specialized resources in the Ln.75 ? Viral specialized resources ?

Please re-frame the sentence for more clarity.

5. Ln.82 too much usage of incompleteness, please re-frame the sentence from "major incompleteness of taxonomically classified" to "major proportion of taxonomically unclassified"

6. CD-Hot tool must be cited, wherever used in the manuscript.

7. I strongly suggest rewriting the manuscript in a concise way, so that less critical sections can be moved to supplementary information. Present manuscript is still in a very rough draft version.

Metagenomics computational analysis related comments:

8. Why authors didn't prefer the traditional and well adopted methodology for metgaenomic assembly using Kraken and Jellyfish for assembling metagenomes and analyzing taxonomic abundance via Phyloseq. ? Why they prefer first de-novo assembly ? If de-novo was crucial, did authors backed-up their results with the above computational workflow for reproducible?

9. Please mention what was the criteria to consider reads under "contaminome" ?

10. Ln507 Please move the last section of the methods to the supplementary methods section.

Authors can concise their methods by making a supplementary method section.

11. some of the tools are cited anywhere (neither in the method section nor in the results/discussion part) in the manuscript.

12. Ln525 what is the rationale behind taking 70% of the length criteria taken?

13. I cant see any link for the catalog? Is it available online ?

This manuscript will be an important contribution to our understanding of the human gut virome. These data support the idea that human gut viromes are highly individual, specific to geographic location (environmental exposure), and in general are not comprised by a “core” set of viruses. I agree with the authors that these and related data are just the beginning of our understanding of viral diversity within the human gut.

The discussion does a good job at identifying important caveats and limitations of the study. A weakness of the study is that it is difficult to reliably make comparisons among SRA data due to the inherent biases introduced from study design/sample collection, all the way through sample processing (both wet and dry lab). The authors acknowledge this limitation and I think it is appropriate to still include these results in the manuscript.

Overall, the authors have generated a massive dataset and performed sound analyses. I believe that this is a good contribution to the field with room to continue to build upon in the future.

Major comments: I put anything under “major comment” that I think needs to be addressed for the reader to properly understand what is being communicated, or places where I feel details may be lacking. I hope that you find these comments to be useful and I believe that it would take overall minor effort to address many of these comments in the text.

1. Line 81: I recommend spending a moment to define viral dark matter for background.
2. Lines 108-110: I think this could be phrased more usefully. What are you trying to distinguish between viral sequences and viral reads here? I.e., perhaps a useful rephrasing could include language referring to the assembled viral sequences that were > 50% complete and in general that 87% of viral reads went into these assemblies.
3. Overall introduction – I think it is important to explain why discovery and characterization of phage is significant, and how this information is useful to the field of virology. Here is a good opportunity to tell us where phage research can be useful.
4. Line 111: I’d like to see more precise language here. Are the “sequences” assemblies, actual raw sequences, or consensus sequences? This critique applies to general precision when describing data throughout the paper.
5. Line 122, Figure 1A: I’d call them putative eukaryotic viruses unless it can be proven

6. Line 128: I think this needs to be stated differently. There are viruses within those families that indeed infect humans, but I think what the authors are saying is that they're not sure if the nearest neighbor to the viruses they detected (that fall within those viral families) actually infect mammals
7. Line 137: in the text or in the methods, I think it would be helpful to state the rationale for using Proteinortho with the 10% cutoff mentioned in line 481.
8. Line 164...does it? Or is it a database limitation? Sampling bias? Temporal difference? I think the results in line 212 are more convincing to this point. Perhaps the HVPC comparison could be excluded or improved to incorporate a broader database...I think it's the comparison of the 2 databases only that makes me scratch my head, right after seeing a comparison to 4-5.
9. Line 192: it seems like a straightforward conclusion here would be that while differing number of phage genomes were detected in patient samples, the overall diversity of the phageome could be captured by the 10 most abundant phage genomes.
10. Can you prove and/or explain how you know that LoVE phage is not a contaminant specific to laboratory reagents or specific collection techniques?
11. Line 342 I think it would add to the discussion to provide more details on similarity before diving into differences in the next thought – recent gut virome studies concluded viromes are individual on which continents, in what populations? Similar demographic/age makeup to this study?
12. Line 351-354 it seems like there are lots of limitations to using the HVPC database. With the caveat that I'm not particularly familiar with it, it might be helpful to the reader to justify why it was included.
13. Line 362: I think this is an informative point
14. I'd like to see a bit more development in the discussion about why it matters that there are more widely-spread phages common to the human gut than just crAssphage. Examples of questions that popped into my mind as I read this: What can we feasibly use this as a marker for? What future studies can build upon this knowledge? Do you think this pattern is limited to humans?/Would you expect to find LoVE-like phages in other mammals? In the environment?

15. Lines 447-449 could be much clearer. I could not replicate abundance calculation based on how it is worded now. A visual describing the bioinformatic workflow could be helpful here.
16. Line 573: I'm curious why all LoVE phage genomes were not submitted to GenBank (including the ones mined from SRA)

Minor comments: I put anything under "minor comment" that may be stylistic in writing, a typo, or that I personally would change for precision and clarity.

1. Line 62 I would refer to it as the "virome" since you are detecting genomes by your methods, not culturing "virobiota"
2. Revise phrasing line 75-77 to be clearer
3. Line 112: it seems excessive to abbreviate viral clusters. I think it would be acceptable to refer to them as "clusters" as something shorter that avoids an acronym
4. Line 115: I would include the threshold of similarity for the clusters to be used for taxonomic classification in the text
5. Line 135: I think it's unclear what the (> x %) refer to
6. Line 137: I would stick with the term orthologous groups, as the abbreviation "OGs" might be interpreted as a joke in US culture.
7. Line 159 define HVPC database
8. Line 195 richness as defined by what calculation? I'd put it in the text
9. Lines 218/19, 306, possibly other places: the data are predicted to encode certain proteins. It's not functionally proven through this work, and so I think it is important to use language indicating "the **predicted** presence of," etc.
10. Line 259: HCST isn't defined until line 524...define earlier
11. Line 262: would be more appropriate to describe this as a crAss-like phage in this heading
12. Line 281: without personally digging into the SRA metadata, I question how the demographics are defined. Are the studies providing metadata self-identified based on ancestry, residence within the past year, etc? Are you pulling this based on location of the publishing institution? If uncertain, it might be more precise to refer to these

observations as “samples collected from healthy patients in [geographic region]” rather than publish an association that might be flawed due to incomplete or partially accurate SRA metadata.

13. Typo line 287 where 6.3% is recorded twice
14. Line 319 missing a space before (Fig. 5D)
15. Line 502: state the year samples were collected/processed for HVPC
16. Line 527-529 should be reworded for clarity.
17. Supplementary Table 2 looks like it's describing assemblies not sequences

This paper describes a new virome database named DEVOC, used here to characterize the healthy gut virome of 91 children, adolescent and adults. This effort is currently highly needed in the community, and this paper is making a very good contribution. I took genuine pleasure in reading it, although some modifications should be made.

Major comments

- 1) My main concern is about the availability of the data and the bioinformatic pipelines used. As I had not access to it, I have no possibilities to verify the results and the pipelines on my own. I would have been pleased with a preprint submitted to biorxiv for example, as well as access to at least the bioinformatic pipeline. I strongly recommend the authors to share the bioinformatic pipeline used in a git repository before publication (an after, of course).
- 2) My second concern is about the abstract. I think it could be modified to present this very nice study a little bit more clearly; it can be improved.
- 3) Finally, it would have been nice to have 16S amplicon sequencing data or shotgun metagenomic data associated with the study (i.e., the microbiota analysis associated). I am not asking for it, but it would certainly have added a plus-value to the paper.

Minor comments

L36-37: "The DEVoC was used to characterize 91 healthy DEVoC gut viromes": unclear. Is it 91 healthy gut viromes taken from the samples used for the database itself? Please clarify this sentence. Also, how did you determine that the individuals are healthy?

L45-46: Why did you focus on the LOVEphage? How did you select the public datasets? How is it related to your DEVOC database? Please explain a little more here and clarify. As present, it seems to be explained/mentioned in the "importance" section of the abstract, but I think it should be explained/clarify here, otherwise it is not clear why you focus on these phages.

L47: I am not sure that "we have contributed" is the good verb to use here? Maybe "we have added".

L60-67: some references are in square brackets [] while others are in parenthesis (). Please correct it. Furthermore, there seem to be a problem in the numbering of the references, at least at the beginning of the introduction (it starts with 1, goes to 5, go back to 4 etc.)

L99: please replace "In particular, a previously undescribed PG, we named LoVEphage" by "In particular, a previously undescribed PG that we named LoVEphage"

L109-110: How did you address the incompleteness issue?

L115-116: 1.4% is very low, it would be nice to have few words about why it is so low and how to deal with it.

L 125: “plant infecting” it is probably due to the alimentation, maybe your interpretation regarding their provenance should be added.

L137: typo: add a space after the comma.

L140: “(40)using”: add space

L170: precise what “ALD” is.

L239: you could have described these 39 highly prevalent PGs in the previous paragraphs, the transition here is a little bit rough

L287: please fully name the disease before to use “IBD” and “CRC”

L313: “18 additional complete LoVEphage-like genomes were reconstructed from the DEVoC and SRA viromes.” How many from DEVoC, how many from SRA?

L 363: “genome level – similar to previous studies (4).” I would rather say “in line with”

L 367: “Gregory et al.” please add the year in the citation.

L379: please precise the notion of core phages, it’s not clear to me here.

L423: “percentile, together with 50 age- and sex-matched healthy controls²⁰”. Citation should be (20) here, please homogenize.

L444: “Raw reads were processed as described by L. Beller et al. (submitted for publication).” It is not enough. I strongly suggest you share the pipelines used for this analysis. Also, some more rational for the thresholds used would help.

L570-574: Same comment as for major concern 1)

L 896: a parenthesis is missing.

In the present manuscript, Espen and Bak et.al. developed a comprehensive database such as Danish Enteric Virome Catalog (DEVoC) by utilizing the human gut virome studies from 204 Danish subjects. They were able to identify the 12,986 non-redundant viral genome sequences encoding 190,029 viral genes. They performed the comparable evaluation of the viral genomes using other virome databases to reveal the unique virome composition in Danish population. Also they showed the utilization of the DEVoc by performing the characterization of healthy gut viromes from Danish people with varying ages, where they were able to dissect the phages prevalent in pediatric versus adult healthy subjects. Further, They also investigated the overall prevalent of phage genomes (PGs) by utilizing 1,880 gut virome samples of 27 studies from across the world. Finally, they observed the high prevalence of crAss-like phages affecting the Bacteroides dorei bacterial species.

Certainly, the work conducted has importance keeping in the view of current growing statistics of gut virome studies and to the extent they usually went underanalyzed to reveal the role of viruses and phages in the disease and also discovering novel phages. However, I do have some major suggestions regarding the manuscript in its present form to improve it to greater extent before it gets published.

Please find the section wise suggestions enlisted below

Manuscript related comments:

- 1. Abstract: I strongly urge the authors to restructure the abstract as its not comprehending the need and importance of the work done.
I) e.g. ln32 in abstract seems incomplete to convey its meaning
II) ln35 we found that
III) ln47 please remove have and change sentence to “we contributed numerous previousl..” and please remove all have in the manuscript. Please consider the English writing and Grammar errors throughout the manuscript.*
- 2. Introduction is not comprehending the role of Human gut Microbiome in Disease. Authors need to add some background history to increase the importance of their conducted work.*
- 3. Introduction: Some key references are missing to emphasize the importance or rationale of the sentence, while some sentences don't have citations at all. See:
I) ln62 only one reference, while there are many key references shedding light on the human virome studies in the disease mechanism, here are some references/PMIDs: 32718358, 32200373, 26489866. A recent one on Inflammatory Bowel disease (IBD) is here: Whole-virome analysis sheds light on viral dark matter in inflammatory bowel disease. Cell Host Microbe. 2019 and many more as cancer immunotherapy. Authors could anchor great importance of their work in introduction by putting forward a well described background and role of virome sequencing in the health; be it gut, skin Microbiome and then mention the need of virome in Danish gut cohorts.
II) Ln. 63, 69, 73 and 74, Ln.81 references are missing,*
- 4. What does authors meant by specialized resources in the Ln.75 ? Viral specialized resources ? Please re-frame the sentence for more clarity.*

5. Ln.82 too much usage of incompleteness, please re-frame the sentence from “major incompleteness of taxonomically classified” to “major proportion of taxonomically **unclassified**”
6. CD-Hot tool must be cited, wherever used in the manuscript.
7. I strongly suggest to rewrite the manuscript in a concise way, so that less critical sections can be moved to supplementary information. Present manuscript is still in a very rough draft version.

Metagenomics computational analysis related comments:

8. Why authors didn't prefer the traditional and well adopted methodology for metagenomic assembly using Kraken and Jellyfish for assembling metagenomes and analyzing taxonomic abundance via Phyloseq. ? Why they prefer first de-novo assembly ? If de-novo was crucial, did authors backed-up their results with the above computational workflow for reproducible ?
9. Please mention what was the criteria to consider reads under “contaminome” ?
10. Ln507 Please move the last section of the methods to supplementary methods section. Authors can concise their methods by making a supplementary method section.
11. some of the tools are cited anywhere (neither in the method section nor in the results/discussion part) in the manuscript.
12. Ln525 what is the rationale behind taking 70% of the length criteria taken ?
13. I cant see any link for the catalog ? Is it available online ?

We would like to thank all four reviewers for their time to read our manuscript and appreciate their constructive comments. We hope that we addressed them adequately in the following sections.

Reviewer 1

This manuscript will be an important contribution to our understanding of the human gut virome. These data support the idea that human gut viromes are highly individual, specific to geographic location (environmental exposure), and in general are not comprised by a “core” set of viruses. I agree with the authors that these and related data are just the beginning of our understanding of viral diversity within the human gut.

The discussion does a good job at identifying important caveats and limitations of the study. A weakness of the study is that it is difficult to reliably make comparisons among SRA data due to the inherent biases introduced from study design/sample collection, all the way through sample processing (both wet and dry lab). The authors acknowledge this limitation and I think it is appropriate to still include these results in the manuscript.

Overall, the authors have generated a massive dataset and performed sound analyses. I believe that this is a good contribution to the field with room to continue to build upon in the future.

Major comments

I put anything under “major comment” that I think needs to be addressed for the reader to properly understand what is being communicated, or places where I feel details may be lacking. I hope that you find these comments to be useful and I believe that it would take overall minor effort to address many of these comments in the text.

1. Line 81: I recommend spending a moment to define viral dark matter for background.

We defined viral dark matter in the Background section.

We replaced lines 80-81:

“However, despite these developments, a large fraction of human gut virome studies still ends up with a significant amount of “viral dark matter””

With (lines 118-120):

“However, despite these developments, a large fraction of sequences originating from human gut virome studies cannot be identified as viral because they are not present in databases and are therefore called “viral dark matter”.”

2. Lines 108-110: I think this could be phrased more usefully. What are you trying to distinguish between viral sequences and viral reads here? I.e., perhaps a useful rephrasing could include language referring to the assembled viral sequences that were > 50% complete and in general that 87% of viral reads went into these assemblies.

We replaced lines 108-110:

“The viral sequences constituting the DEVoC ranged in size from 1 kb to 191 kb (N50: 16 kb; L50: 1,463 scaffolds) and while CheckV (36) estimated that only 1,867 viral sequences (14.4%) were more than 50% complete, these sequences represented 87.4% of the total amount of viral reads.”

With (lines 156-159):

“The viral scaffolds constituting the DEVoC ranged in size from 1 kb to 191 kb (N50: 16 kb; L50: 1,463 scaffolds) of which 1,867 viral sequences (14.4%) were more than 50% complete as estimated by CheckV (36). This small subset of viral sequences however dominates these Danish fecal viromes as they represented 87.4% of the total amount of viral reads.”

3. Overall introduction – I think it is important to explain why discovery and characterization of phage is significant, and how this information is useful to the field of virology. Here is a good opportunity to tell us where phage research can be useful.

Thank you for this suggestion. We rewrote the first section of the Introduction and changed the following lines (lines 66 – 69) in the Introduction specifically to address your comment to better convey the importance of phage research:

“Since bacteria and viruses are the two most abundant components of the human gut microbiota, shedding more light on the virobiota, and their collective genomes referred to as the virome, will pave the way to unraveling complex interactions within the gut microbiota and their effect on the human host.”

Into (lines 93-106)

“The close interplay between phages and bacteria, which are already implicated in numerous diseases, combined with the ability of gut viruses to directly interact with the human host (Ogilvie & Jones 2015, Tetz & Tetz 2017), led to gut viruses gaining more interest as potential disease biomarkers (Nakutsu et al 2018) and treatments for disease (Duan et al 2020, Ott et al 2017). It is therefore important to shed more light on the virobiota, and their collective genomes referred to as the virome, as this will pave the way to unravel complex interactions within the gut microbiota and their effect on the human host (Sutton & Hill 2019).”

4. Line 111: I'd like to see more precise language here. Are the “sequences” assemblies, actual raw sequences, or consensus sequences? This critique applies to general precision when describing data throughout the paper.

We changed “sequences” into “scaffolds” throughout the paper.

5. Line 122, Figure 1A: I'd call them putative eukaryotic viruses unless it can be proven

We changed this accordingly (revised manuscript line 131 & 133)

6. Line 128: I think this needs to be stated differently. There are viruses within those families that indeed infect humans, but I think what the authors are saying is that they're not sure if the nearest neighbor to the viruses they detected (that fall within those viral families) actually infect mammals.

As pointed out by Reviewer #2 it could indeed be that the closest neighbor does not necessarily infect the same mammal as the sequence detected in our samples, hence we opted for “mammal” and not “human” infecting. Our main reason for writing “potentially infecting mammals” was that for some of these viruses we know they infect mammals (e.g. Enterovirus, Sapovirus), while for others it has been hypothesized they infect mammals (e.g. *Anelloviridae*, *Genomoviridae*) but evidence is still lacking. To clarify this, we replaced line 128: “viral families potentially infecting mammals” with “viral families that are known or hypothesized to infect mammals” (line 192-193)

7. Line 137: in the text or in the methods, I think it would be helpful to state the rationale for using Proteinortho with the 10% cutoff mentioned in line 481.

We clarified the following statement in the Methods section (lines 480-481):
“The annotation given to at least 10% of the OG members was assigned to the OG.”

With (lines 645-649):

“The annotation(s) given to at least 10% of the protein members of a specific OG was assigned that OG of interest (manually, to overcome differences in spelling, capitalization and abbreviations as well as synonyms of the same protein due to the use of different databases). This 10% thresholds was introduced to avoid spurious annotations of OGs.”

8. Line 164...does it? Or is it a database limitation? Sampling bias? Temporal difference? I think the results in line 212 are more convincing to this point. Perhaps the HVPC comparison could be excluded or improved to incorporate a broader database...I think it's the comparison of the 2 databases only that makes me scratch my head, right after seeing a comparison to 4-5.

As comparison to the HVPC is not very informative (due to its non-gut specificity and no viral selection), we decided to remove this section.

Therefore, Fig. 1D, lines 159 – 166 (Results), lines 351 – 354 (Discussion), lines 501 – 506 (Methods), lines 902-905 (Figure Legends) and all other references to this analysis are removed

And the following statement was added to the discussion (line 351):

“As there is currently no database cataloguing the proteins encoded by gut viral genomes, the DEVoC encoded proteins could serve as a starting point to study the functional capacity of human gut viromes”.

9. Line 192: it seems like a straightforward conclusion here would be that while differing number of phage genomes were detected in patient samples, the overall diversity of the phageome could be captured by the 10 most abundant phage genomes.

We extended lines 190 – 192:

“The most abundant PG within each subject recruited between 0.24% and 83.2% of the phage reads (median: 30.4%; IQR: 20.7% - 44.0%; **Fig. 2B**), while the 10 most abundant PGs represented the majority of the phage reads in most subjects (median: 82.4%; IQR: 69.2% - 89.0%; range: 0.83% – 99.5%; **Fig. 2B**).”

With: “This suggests that the overall diversity of the phageome can be captured by the 10 most abundant PGs in most samples.”

10. Can you prove and/or explain how you know that LoVE phage is not a contaminant specific to laboratory reagents or specific collection techniques?

The fact that we find the LoVE phage back in numerous other human gut virome studies from other laboratories using a variety of different wet-lab procedures and kits indicates that is not a contaminant.

11. Line 342 I think it would add to the discussion to provide more details on similarity before diving into differences in the next thought – recent gut virome studies concluded viromes are individual on which continents, in what populations? Similar demographic/age makeup to this study?

We changed line 342-346:

“Recent human gut virome studies have all concluded gut viromes to be highly individual (8, 10). Individual-differentiating factors likely include geographical origin (46), age (4), diet (7) and health status (47–49). Therefore, it is not unsurprising that the majority of the identified viral genomes and genes were not previously described, indicating that we are only scratching the surface of the viral diversity in the human gut microbiota worldwide (**Fig. 1C**).”

Into (lines 463-475):

“Recent human gut virome studies concluded gut viromes of healthy Western adults (Shkoporov et al. 2019, Moreno-Gallego et al. 2019, Garmaeva et al. 2021) to be highly individual. Individual-differentiating factors likely include geographical origin (Rampelli et al 2017), age (Gregory et al 2020), diet (Minot et al. 2017, Garmaeva et al. 2021) and health status (Norman et al. 2015, Zhao et al. 2017, Clooney et al. 2019, Lang et al. 2020, Jiang et al. 2020). Hence, our findings of large individuality in the gut virome composition in healthy Danish adults is expected. The virome composition of healthy children (> 3 years) and adolescents has not been studied before, but expected to show similar subject specificity, since gut virome individuality has also been observed in infants (Maqsood 2019). Due to virome individuality, it is not unsurprising that the majority of the identified viral genomes were not previously described, indicating that we are only scratching the surface of the viral diversity in the human gut microbiota worldwide (**Fig. 1C**).”

12. Line 351-354 it seems like there are lots of limitations to using the HVPC database. With the caveat that I’m not particularly familiar with it, it might be helpful to the reader to justify why it was included.

We removed to comparison with HVPC database. See comment #8.

13. Line 362: I think this is an informative point

We appreciate this comment.

14. I'd like to see a bit more development in the discussion about why it matters that there are more widely-spread phages common to the human gut than just crAssphage. Examples of questions that popped into my mind as I read this: What can we feasibly use this as a marker for? What future studies can build upon this knowledge? Do you think this pattern is limited to humans?/Would you expect to find LoVE-like phages in other mammals? In the environment?

We added the following lines to the Discussion (lines 556-560):

“Future studies should experimentally determine the host range and morphology of the LoVEphage, as well as explore the broad genetic diversity in the human population to uncover the potential associations of variants with specific diseases. It could be worthwhile to also investigate whether this phage is specific to humans, or is also found in non-human primates and other mammals as is the case with the crAss-like phages (Edwards et al 2019, Li et al. 2021).

15. Lines 447-449 could be much clearer. I could not replicate abundance calculation based on how it is worded now. A visual describing the bioinformatic workflow could be helpful here.

We replaced lines 447-449:

“Abundances were determined by mapping the quality-filtered reads to a subset of the cluster representatives containing only those of clusters containing at least one scaffold from that sample.”

With (lines 599-607):

“Instead of calculating abundances by mapping quality-filtered reads to the complete set of non-redundant scaffolds, reads were only mapped to the representatives of the clusters containing a scaffold from that sample to avoid false positive detection of closely related sequences.”

16. Line 573: I'm curious why all LoVE phage genomes were not submitted to GenBank (including the ones mined from SRA)

The additional LoVEphage-like genomes are submitted to GenBank in the meantime, but do not yet have an assigned accession number.

Minor comments

I put anything under “minor comment” that may be stylistic in writing, a typo, or that I personally would change for precision and clarity.

1. Line 62 I would refer to it as the “virome” since you are detecting genomes by your methods, not culturing “virobiota”.

We understand your suggestion however, here we used “virobiota” because we are discussing the role of the human gut viruses (and not their genomes, which would be referred to as “virome”). In this manuscript, we are indeed studying their genomes and hence refer to “virome” onwards when discussing our own results.

2. Revise phrasing line 75-77 to be clearer

We changed line 75-77:

“High viral genomic mutational rates further add to the incompleteness of databases, by creating immense viral genetic diversity (19).”

To (lines 114-115):

“High viral mutation rates cause immense viral genetic diversity, thereby complicating viral identification based on homology to reference genomes (19).”

3. Line 112: it seems excessive to abbreviate viral clusters. I think it would be acceptable to refer to them as “clusters” as something shorter that avoids an acronym.

Viral clusters generated by vConTACT2 are very often referred to as VCs (see Bin Jang et al. 2019, Clooney et al. 2019, Gregory et al. 2020, Garmaeva et al. 2021). Also we use “VCs” to avoid confusion with the “clusters” generated by clustering the sample-specific *de novo* assembled scaffolds to form a non-redundant database (including non-viral clusters).

4. Line 115: I would include the threshold of similarity for the clusters to be used for taxonomic classification in the text

No similarity thresholds are used for taxonomic classification. vConTACT2 uses gene-sharing networks (based on protein clusters) to viral genomes with viral RefSeq genomes (using default parameters). If a RefSeq genome is part of a VC, the classification of that RefSeq genome is transferred to the other VC members up to genus level.

5. Line 135: I think it's unclear what the (> x %) refer to

The percentages refer to the percentage of DEVoC proteins that have these annotations. Percentages are minimum, as numerous different versions (spelling, capitalisations, abbreviations, synonyms) can indicate the same protein and it is impossible to know if we included all variants in the counts.

We removed the percentages to avoid confusion. Also see major comment #7.

6. Line 137: I would stick with the term orthologous groups, as the abbreviation “OGs” might be interpreted as a joke in US culture.

Although we appreciate your concern, we decided to keep this commonly used abbreviation (see Huerta-Cepas et al 2019, Li et al 2014) to save space and hope readers will interpret the abbreviation as we intended.

7. Line 159 define HVPC database

We removed to comparison with HVPC database. See comment #8.

8. Line 195 richness as defined by what calculation? I'd put it in the text

We replaced: "The median eukaryotic viral species richness" by "The median observed eukaryotic viral species richness" (line 294)

9. Lines 218/19, 306, possibly other places: the data are predicted to encode certain proteins. It's not functionally proven through this work, and so I think it is important to use language indicating "the predicted presence of," etc.

Line 134: We changed "with the most common annotations being" into "with the most common predicted annotations being" (line 198)

Line 204: We changed "The five most prevalent OGs included ..." into "The five most prevalent OGs (...) were predicted to encode ..." (line 302-303)

Line 218-219: We changed "PG8 encoded proteins involved in the activation or suppression of the lysogenic cycle" into "PG8 is predicted to encode proteins involved in the activation or suppression of the lysogenic cycle" (line 318-319)

Line 294: We changed "one region encoding structural proteins and proteins involved in host interaction, the other region encoding protein involved in ..." into "one region encoding proteins predicted to be involved in host interaction and phage structure, the other region encoding proteins predicted to be involved in ..." (line 407-408)

Line 306: We changed "The presence of two integrase proteins, a repressor protein, and a prophage protein suggest that the LoVEphage is a temperate phage" into "The predicted presence of two integrase proteins, a repressor protein, and a prophage protein suggest that the LoVEphage is a temperate phage." (line 420-421)

10. Line 259: HCST isn't defined until line 524...define earlier

All disease abbreviations are now defined on lines 258 – 261.

11. Line 262: would be more appropriate to describe this as a crAss-like phage in this heading

We replaced: "A crAssphage and a previously undescribed phage were highly prevalent in healthy Danish subjects and shared across the world"

With (line 369-370):

“A crAss-like phage and a previously undescribed phage were highly prevalent in healthy Danish subjects and shared across the world”

12. Line 281: without personally digging into the SRA metadata, I question how the demographics are defined. Are the studies providing metadata self-identified based on ancestry, residence within the past year, etc? Are you pulling this based on location of the publishing institution? If uncertain, it might be more precise to refer to these observations as “samples collected from healthy patients in [geographic region]” rather than publish an association that might be flawed due to incomplete or partially accurate SRA metadata.

We screened the subject recruitment section from the papers, biosamples/SRA entries and/or associated supplementary information about the demographics. Detailed resident information (duration and city-specific location) and ancestry was only rarely specified, so we indeed used the region of inclusion (e.g. hospital), which was always available and assumed that individuals included in these locations also live(d) there.

We added a clarifying statement in the Methods section (line 541-542):

“The metadata were curated by screening the original articles subject recruitment section, supplementary demographic tables and/or the information associated with the BioSample/SRA entry.”

And in the Results section (line 244 – revised manuscript line 346): “geographical region (continent of sample collection)”

13. Typo line 287 where 6.3% is recorded twice

Removed.

14. Line 319 missing a space before (Fig. 5D)

Added.

15. Line 502: state the year samples were collected/processed for HVPC

We removed to comparison with HVPC database. See comment #8.

16. Line 527-529 should be reworded for clarity.

“The prevalent crAss-like phage genome as clustered with the 249 genomes from the crAss-like phage dataset of Guerin *et al.* (54) using ClusterGenomes (85) at 95% identity over 80% coverage to determine to which proposed genus/subfamily the genome belongs. “

“To determine to which proposed genus/subfamily the prevalent crAss-like phage genome (PG2) belongs, its genome is clustered with the 249 genomes from the crAss-like phage dataset of Guerin *et al.* (54) using ClusterGenomes (85) at 95% identity over 80% coverage. “

17. Supplementary Table 2 looks like it's describing assemblies not sequences

We replaced "sequences" with "scaffolds" throughout the manuscript. See major comment #4.

Reviewer #2

For the article "A previously undescribed highly prevalent phage identified in a Danish enteric virome catalog" submitted to review in mSystems, in general, I have no objections to the methodology proposed by the authors. It seems very accurate and up-to-date to me. However, there are some things that I would like to see modified, clarified or improved in the text.

1. For example, the term "viral individuality" or "degree of viral individuality", I do not think is the best to refer to the specificity of the virobiota of each individual, if that is the case. Certainly, there is a high degree of viral specificity in the gut microbiota, but the term "viral individuality" sounds like something else.

We avoided the use of "viral individuality" and instead used terms such as "interindividual virome diversity" or "individual specificity of the human gut virome", "highly individual gut virome" and "virome individuality" as used by Shkoporov et al (2019) to describe the human gut virome as highly diverse, stable and individual specific.

2. Did the 91 Danish subjects have the same diet or at least a similar diet for a particular period of time? This could have an impact on the determination of the intestinal virome diversity of each one of them.

We agree with reviewer's #2 concern that diet could potentially have an impact on gut virome composition and diversity. Unfortunately, detailed dietary information was not available for these subjects and could hence not be taken into account.

3. It is not clear to me why the authors do mention that 39 of the most frequent phage genomes in the samples are present in more than 10 subjects (> 12%). That is, the authors worked with 91 healthy individuals (46 children/adolescents + 45 adults). Why not define how many of these individuals/subjects had the 39 phage genomes? Were these 39 PGs present in all the 81 remaining subjects?

We apologies for this unclarity. The 39 phage genomes occurred in 11 to 30 of the healthy subjects (i.e. more than 10 subjects – prevalences: 12 – 33%)

In lines 182-183 we stated the following:

"The most prevalent PG was a partial *Skunavirus* genome detected in 33% of the subjects. Only a limited number of PGs (n = 39; 3.4%) was found across more than 10 subjects (**Table S3**)."

And changed this into (lines 270-272)

"The most prevalent PG was a partial *Skunavirus* genome detected in 30 of the subjects (33% prevalence). Including this PG, only 39 PGs (3.4% of all PGs) occurred in more than 10 subjects (> 12% prevalence; **Table S3**)."

4. I understand that statistically, it is significant that PG8 was more abundant in children/adolescents, while PG7 and PG22 were more abundant in adults. However, PGs are

still present (at least detected in 1 subject) albeit in low proportions in its counterpart. This could be a problem due to the n of samples that authors use.

We understand the reviewer's concern, but at the same time we believe that our conclusions are supported by our data. Sample sizes are taken into account in the Chi² test and the test was significant, **also after correction for multiple testing** using the Bonferroni method which is rather stringent. Moreover, we never claim that these PGs are specific to a certain age group, but rather associated with/more prevalent in a certain age group. Furthermore, in comparisons to other human gut virome papers that compare prevalence/abundance of certain viruses/viral groups between disease groups or age groups, our sample sizes are on the larger side (cfr. Draper et al. 2018: n = 3 vs. n = 11, Fernandes et al. 2019: n = 12 vs. n = 5 vs. n = 7, Hannigan et al. 2019: n = 30 vs. n = 30 vs. n = 30, Maqsood et al. 2019: n = 28 vs. n = 66, Zhao et al. 2017: n = 11 vs. n = 11, Lang et al. 2020: n = 29 vs. n = 44, n = 9 vs. n = 13, Jiang et al. 2020: n = 89 vs. n = 36 vs. n = 17)

5. Do you have how to discuss why this difference in abundance of phages (or phage genomes) could occur in children/adolescents vs adults?

What relevance could have the fact of found differences in the abundance of PGs between children/adolescents vs adults, and how does this correlate with the title of this article?

We added the following lines to the Discussion (line 518-524):

"Moreover, while age does not seem to affect overall diversity of the gut virome, age seems to affect the presence of certain viruses. The association of specific phages with age group might be linked to associations of their respective bacterial hosts and could consequently influence the interaction of the gut microbiota with the human host, affecting human host metabolism and immune response."

6. Why in Figure 4 you separate the group of adults (45 subjects) into adults (34 subjects) and the elderly (11 subjects)? Just to compare with SRA viromes? Why not follow the same nomenclature that they used throughout the study? This separation, although it serves to compare the SRAs, would also affect the heat map of the 39 PGs.

During the initial analysis, we extensively discussed how we should use the "age" variable in our study.

At first, we opted to use age as a continuous variable (for diversity measures, age-association of individual PGs, ...) . However, since we do not have subjects in the 19 – 39 year old range, we decided it was best if we used the age groups naturally formed by using two cohorts (pediatric and adult), as age was normally distributed within those two cohorts separately.

As exact age information was often not available for the SRA subjects, we decided to use age groups as well (instead of "continuous age") to include as many SRA subjects as possible in the comparison. However, as some SRA studies specifically included elderly patients up to 99 years old (Stockdale et al. 2018), we decided to split the adult group into non-elderly adults and elderly to make

our comparison more sensitive to changes (especially since we had a relatively high number of healthy elderly subjects).

After that we considered splitting up our own adult cohort into non-elderly adults and elderly, but due to close age proximity of both groups (especially with regard to the large gap with the pediatric cohort) and due to the limited number of elderly subjects in comparison to non-elderly adults in our own cohort, we decided against this.

We hope you understand our reasoning. However, we do understand your concern that it is not logical to split our own adult cohort into non-elderly adults and elderly adults in Fig. 4A (without discussing this) and have therefore opted to group them both into one column in Fig. 4A with "Adults (incl. elderly) n = 45" as column name.

7. Does the order (ascending or descending) of the graphical representation of the 19 genomes of LoVE phage (Figure 5D) follow the same order of appearing (ascending or descending, respectively) of these genomes in the phylogenetic tree in Figure 5C? If so, why not merge both images into one?

Yes, the order indeed matches. We removed the "D" in the figure 5 and clarified the legend.

We replaced lines 946-949...951-952: *"(C) Maximum likelihood trees of concatenated protein alignments (n = 61) of 19 LoVEphage-like genomes recovered from Danish subjects and SRA viromes. Only bootstrap values higher than 70 are shown. Tip symbol and color are indicative of respectively, the age group and country of origin of the individual from which the LoVEphage-like genome was assembled. (D) Genome structure of the 19 LoVEphage-like genomes shown in panel C."*

With: *"(C) Maximum likelihood trees of concatenated protein alignments (n = 61) of 19 LoVEphage-like genomes recovered from Danish subjects and SRA viromes. Only bootstrap values higher than 70 are shown. Tip symbol and color are indicative of respectively, the age group and country of origin of the individual from which the LoVEphage-like genome was assembled. Full genome structures of the 19 LoVEphage-like genomes are visualized next to the phylogenetic tree. ..."*

We also replaced all references to Fig5D by Fig 5C

8. Since authors found a high incidence of phage sequences belonging to Firmicutes and Bacteroidetes as gut hosts, I would suggest they incorporate this article to discuss their results:

Koliada, A., Moseiko, V., Romanenko, M. et al. Sex differences in the phylum - level human gut microbiota composition. BMC Microbiol 21, 131 (2021). <https://doi.org/10.1186/s12866-021-02198-y>

Thank you for suggesting this interesting paper. We added this reference to line 339-341: "DEVoC phages (*Caudovirales* and *Petitvirales* orders) and their

bacterial hosts (*Firmicutes* and *Bacteroidetes* phyla) have all been commonly described in the human gut.”

9. In general, I found a very descriptive article. I would like to see a juicier discussion.

We added several additional points to the Discussion section.

- Lines 455-460: “Although less stringent taxonomical classification approaches could increase the number of phage genomes with assigned taxonomy, a large fraction of phage scaffolds would remain unclassified nonetheless, hampering potential subsequent analyses at family/genus level. Hence, further analyses were conducted at individual scaffold level, thereby also avoiding the results to become outdated due to the constantly evolving phage taxonomy.”
- Lines 469-472: “Hence, our findings of large individuality of the gut virome in healthy Danish adults is expected. The virome composition of healthy children (> 3 years) and adolescents has not been studied before, but is expected to show similar subject specificity since gut virome individuality has also been observed in infants.”
- Lines 518-524: “Moreover, while age does not seem to affect overall diversity of the gut virome, age seems to affect the presence of certain viruses. The association of specific phages with age group might be linked to the gut microbiota with the human host, affecting human host metabolism and immune response.”
- Lines 556-560: “Future studies should experimentally determine the host range and morphology of the LoVEphage, as well as their broad genetic diversity in the general population to uncover the potential associations of variants with disease. It could be worthwhile to also investigate whether this phage is specific to humans, or is also found in non-human primates and other mammals as is the case for crAss-like phages (74, 75)”

10. The initials of the authors should be checked in the "Author contributions" section.

Thank you for pointing this out. We changed “LCC” into “LC” on line 782.

Reviewer #3

This paper describes a new virome database named DEVOC, used here to characterize the healthy gut virome of 91 children, adolescent and adults. This effort is currently highly needed in the community, and this paper is making a very good contribution. I took genuine pleasure in reading it, although some modifications should be made.

Major comments

1) My main concern is about the availability of the data and the bioinformatic pipelines used. As I had not access to it, I have no possibilities to verify the results and the pipelines on my own. I would have been pleased with a preprint submitted to biorxiv for example, as well as access to at least the bioinformatic pipeline. I strongly recommend the authors to share the bioinformatic pipeline used in a git repository before publication (an after, of course).

We appreciate your concern regarding data availability and agree that is important for reproducibility..

The raw sequencing reads are deposited to SRA under BioProject PRJNA723467 (pediatric cohort) and PRJNA722819 (adult cohort) and have the following reviewer links:

- <https://dataview.ncbi.nlm.nih.gov/object/PRJNA723467?reviewer=8dquji7th42plu68ejrbpfe4ev>
- <https://dataview.ncbi.nlm.nih.gov/object/PRJNA722819?reviewer=361ikob3off4do653upj5ilahm>

The BioProjectsIDs have been added to the “Data availability” statement in the Methods section at line 571 (revised manuscript lines 600-601) and will be made publicly available after publication.

A preprint of the paper is available at ResearchSquare (<https://www.researchsquare.com/article/rs-273865/v1>).

We tried to explain the bioinformatic pipeline as detailed as possible in the Methods section for readers to be able to replicate our findings. Scripts are made available on github

(<https://github.com/Matthijnssenslab/ViromeCatalogue>).

Datasets (including DEVoC, DEVoC genes, annotations and counts are available at <https://doi.org/10.5281/zenodo.5173012>

2) My second concern is about the abstract. I think it could be modified to present this very nice study a little bit more clearly; it can be improved.

We rewrote the abstract and hope that we thereby improved its clarity.

3) Finally, it would have been nice to have 16S amplicon sequencing data or shotgun metagenomic data associated with the study (i.e., the microbiota analysis associated). I am not asking for it, but it would certainly have added a plus-value to the paper.

Shotgun metagenomic sequencing is available for the same samples (and has been used for host prediction as indicated on lines 467 – 469). However, analysis of the bacterial gut microbiome composition was not part of the aim of this study and will be part of a future paper.

Minor comments

L36-37: "The DEVoC was used to characterize 91 healthy DEVoC gut viromes": unclear. Is it 91 healthy gut viromes taken from the samples used for the database itself? Please clarify this sentence. Also, how did you determine that the individuals are healthy?

We clarified the statement replacing lines 36 – 37 with the following (lines 35-36):

"The DEVoC was used to compare 91 healthy gut viromes that were used to create the DEVoC."

The individuals were determined to be healthy based on not taking any medication at time of inclusion and not taking antibiotics at least 6 months prior to inclusion, the absence of any chronic disease and limited alcohol intake.

L45-46: Why did you focus on the LOVEphage? How did you select the public datasets? How is it related to your DEVoC database? Please explain a little more here and clarify. As present, it seems to be explained/mentioned in the "importance" section of the abstract, but I think it should be explained/clarify here, otherwise it is not clear why you focus on these phages.

We clarified lines 45 – 46:

"A *de novo* assembly of selected public datasets generated an additional 18 circular LOVEphage-like genomes (67.9-72.4 kb)."

By replacing them with the following (lines 48-50):

"Due to the LOVEphage's high prevalence and novelty, public datasets in which the LOVEphage was detected were *de novo* assembled, resulting in an additional 18 circular LOVEphage-like genomes (67.0 – 72.4 kb)."

Public datasets were selected based on a PubMed search in December 2019 searching for "human gut/fecal/enteric viromes". Studies where raw sequencing reads and/or metadata (subjectID, health status, age group) were unavailable were excluded as well as studies using targeted sequencing or not using viral enrichment. These datasets are all unrelated to the DEVoC database as the DEVoC database only included newly sequenced samples.

We added the following information to the Methods section:

Lines 690-694: "Studies were selected based on a PubMed search in December 2019 searching for "human gut/fecal/enteric viromes". Studies using targeted sequencing or not using viral enrichment were excluded as well as studies for which raw sequencing reads and/or metadata (subjectID, health status, age group, geographic region of inclusion) were unavailable."

L47: I am not sure that "we have contributed" is the good verb to use here? Maybe "we have added".

Thank you for this suggestion. We changed “contributed” with “added”.

L60-67: some references are in square brackets [] while others are in parenthesis (). Please correct it. Furthermore, there seem to be a problem in the numbering of the references, at least at the beginning of the introduction (it starts with 1, goes to 5, go back to 4 etc.)

Thank you for pointing this out this reference issue! We corrected it.

L99: please replace "In particular, a previously undescribed PG, we named LoVEphage" by "In particular, a previously undescribed PG that we named LoVEphage"

We replaced this as requested.

L109-110: How did you address the incompleteness issue?

At the moment, there is unfortunately no good tool for phage genome binning. Most phages do not occur in enough samples to use an abundance correlation method to group fragments from the same genome. To avoid including multiple fragments of the same genome in further diversity analysis (and subsequently overestimating for example richness estimates), we continue only with the fragments estimated to be at least 50% complete as stated in lines 172 – 176: “As multiple fragments from the same genome can hamper phage community-level analysis when they are treated as separate viruses, we restricted the analysis to phage sequences that represented more than 50% of a genome as determined by CheckV (36) (hereafter referred to as Phage Genomes; PGs). This allows us to limit the analysis to a maximum of one fragment for any given genome.”

We believe this is the most optimal approach at this time, as this fraction of scaffolds dominate most gut viromes and represent 87.4% of all viral reads as stated in line 109.

L115-116: 1.4% is very low, it would be nice to have few words about why it is so low and how to deal with it.

1.4% is indeed very low (especially in comparison to other human gut virome studies) and this can be partially explained by

- 1) The rather stringent approach to assign taxonomy (vConTACT2 clustering with RefSeq genomes). Other papers often rely on similarity of a single protein to a reference protein to assign taxonomy. We preferred to be conservative and only assign taxonomy to a limited set of scaffolds and be confident about these classifications.

- 2) The predominance of short and incomplete phage genome fragments in the DEVoC hamper classification of phage genomes using vConTACT2 as it relies on gene sharing networks. Other studies often set their genome length cut-offs much higher (at 3 to 5 kb), but we opted to go for a lower cut-off of 1 kb to include short, but (near-)complete (segmented) viral

genomes such as *Anelloviridae*, *Circoviridae*, *Microviridae*, *Partitiviridae*, etc.

As phage taxonomy is, even with less stringent approaches, not available for a large fraction of the phage genomes and major updates to shift from morphology-based to sequence-based classifications are frequent, we believe that conducting analysis at sequence level is the most ideal at the moment. Moreover, with 11 out of the 39 PGs having taxonomical classifications and the 176 phage genomes with assigned taxonomy (1.4%) taking up 9.8% of the phage reads, many of the more abundant/prevalent phage genomes could be classified.

We added a few lines in the Discussion section (lines 455-460):
“Although less stringent taxonomical classification approaches could increase the number of phage genomes with assigned taxonomy, a large fraction of phage scaffolds would remain unclassified nonetheless, hampering potential subsequent analyses at family/genus level. Hence, further analyses were conducted at individual scaffold level, thereby also avoiding the results to become outdated due to the constantly evolving phage taxonomy”.

L 125: "plant infecting" it is probably due to the alimentation, maybe your interpretation regarding their provenance should be added.

We added the following to line 174-190: “probably originating from the diet”

L137: typo: add a space after the comma.

Corrected.

L140: "(40)using": add space

Corrected.

L170: precise what "ALD" is.

Done.

L239: you could have described these 39 highly prevalent PGs in the previous paragraphs, the transition here is a little bit rough

We added the following statement to line 185: “This subset of 39 highly prevalent PGs will be further looked into in the next sections.”
And changed “the 39 most prevalent PGs” into “the subset of the 39 most prevalent PGs” at line 315.
and changed “We assessed whether the 39 highly prevalent PGs in the healthy Danish subset could be recovered worldwide, across age groups and diseases”

into “We further examined whether the subset of 39 highly prevalent PGs earlier defined in the healthy Danish subjects could be recovered from a diverse worldwide population, across age groups and diseases.”
to hopefully make the transitions smoother and clearly indicate that this is always the same subset of 39 PGs.

L287: please fully name the disease before to use "IBD" and "CRC"

All disease related abbreviations in this section are now defined before use.

L313: "18 additional complete LoVEphage-like genomes were reconstructed from the DEVoC and SRA viromes." How many from DEVoC, how many from SRA?

We added 3 and 15, respectively.

L 363: "genome level - similar to previous studies (4)." I would rather say "in line with"

Changed.

L 367: "Gregory et al." please add the year in the citation.

Added.

L379: please precise the notion of core phages, it's not clear to me here.

We replaced lines 378 – 380:

“The presence of age-associated PGs may indicate that some more common (or even core) phages might exist in smaller, more homogeneous, populations, although core phages do not exist for the general healthy human population.”

With:

“The presence of age-associated PGs may indicate that some more common (prevalence between 20 – 50%) or even core (prevalence higher than 50%) phages (Manrique et al 2016) might exist in smaller, more homogeneous, populations, although core phages do not exist for the general healthy human population.”

L423: "percentile, together with 50 age- and sex-matched healthy controls²⁰". Citation should be (20) here, please homogenize.

Corrected.

L444: "Raw reads were processed as described by L. Beller et al. (submitted for publication)." It is not enough. I strongly suggest you share the pipelines used for this analysis.

Also, some more rational for the thresholds used would help.

See major concern #1.

The statement on lines 446 – 447: “Quality filtered reads were *de novo* assembled and all scaffolds longer than 1 kb were clustered at 95% identity over 80% coverage to remove redundancy” was extended with “in line with Roux et al. 2017”.

Thresholds for breadth of coverage for scaffolds presence are commonly between 70 and 75% (Shkoporov et al 2019, Gregory et al 2020).

The statement on lines 450 – 451: “Scaffolds representing less than 0.00001% of the total amount of mapped reads across all samples were removed” was extended with “to reduce background noise”.

L570-574: Same comment as for major concern 1)

See major concern #1.

L 896: a parenthesis is missing.

Added.

Reviewer #4

In the present manuscript, Espen and Bak et.al. developed a comprehensive database such as Danish Enteric Virome Catalog (DEVoC) by utilizing the human gut virome studies from 204 Danish subjects. They were able to identify the 12,986 non-redundant viral genome sequences encoding 190,029 viral genes. They performed the comparable evaluation of the viral genomes using other virome databases to reveal the unique virome composition in Danish population. Also they showed the utilization of the DEVoc by performing the characterization of healthy gut viromes from Danish people with varying ages, where they were able to dissect the phages prevalent in pediatric versus adult healthy subjects. Further, They also investigated the overall prevalent of phage genomes (PGs) by utilizing 1,880 gut virome samples of 27 studies from across the world. Finally, they observed the high prevalence of crAss-like phages affecting the *Bacteroides dorei* bacterial species.

Certainly, the work conducted has importance keeping in the view of current growing statistics of gut virome studies and to the extent they usually went underanalyzed to reveal the role of viruses and phages in the disease and also discovering novel phages. However, I do have some major suggestions regarding the manuscript in its present form to improve it to greater extent before it gets published.

Please find the section wise suggestions enlisted below

Manuscript related comments:

1. Abstract: I strongly urge the authors to restructure the abstract as its not comprehending the need and importance of the work done.

We rewrote the abstract and hope that the importance section conveys the need and importance of the work done.

I) e.g. ln32 in the abstract seems incomplete to convey its meaning

As we do not have the space to dig deeper into these challenges (but do so in the introduction), we removed "as many wet lab and bioinformatic challenges remain" from line 32.

II) ln35 we found that

We added "we found that" to the sentence.

III) ln47 please remove have and change sentence to "we contributed numerous previously.." and please remove all have in the manuscript. Please consider the English writing and Grammar errors throughout the manuscript.

We carefully went through the entire manuscript and removed "have" where appropriate.

2. Introduction is not comprehending the role of the Human gut Microbiome in Disease.

Authors need to add some background history to increase the importance of their conducted work.

We rewrote the first part of the Introduction to give more insights into the role of gut microbiome (including gut virome) in human health and disease and why studying the gut virome in health and disease is important.

Specifically we added

- “Both structural and functional imbalances of the gut bacteria, called dysbiosis, have been associated with diseases such as obesity (Gerard et al. 2016), diabetes (Gurung et al 2020, Zheng et al. 2018), inflammatory bowel disease (IBD) (Nishida et al. 2017), cancer (Cheng et al. 2020) and neurological diseases (Ma et al. 2019).” at lines 84-86.
- “At the same time, research on human gut viruses, collectively called gut virobiota, is still in its infancy (Garcia-Lopez et al. 2019), although recent studies have demonstrated associations with IBD (Liang et al. 2020, Clooney et al. 2020), diabetes (Chen et al. 2021, Vehik et al. 2019) liver disease (Lang et al. 2020, Jiang et al. 2020) and cancer (Nakatsu et al. 2018).” at lines 86-88.
- “The close interplay between phages and bacteria, which are already implicated in numerous diseases, combined with the ability of gut viruses to directly interact with the human host (Ogilvie et al. 2015, Tetz et al. 2018), led to gut viruses gaining more interest as potential disease biomarkers (Nakatsu et al. 2018) and treatments for disease (Duan et al. 2019, Ott et al 2017).” at lines 93-96.

3. Introduction: Some key references are missing to emphasize the importance or rationale of the sentence, while some sentences don't have citations at all. See:

l) In62 only one reference, while there are many key references shedding light on the human virome studies in the disease mechanism, here are some references/PMIDs: 32718358, 32200373, 26489866. A recent one on Inflammatory Bowel disease (IBD) is here: Whole-virome analysis sheds light on viral dark matter in inflammatory bowel disease. Cell Host Microbe. 2019 and many more as cancer immunotherapy. Authors could anchor the great importance of their work in the introduction by putting forward a well-described background and role of virome sequencing in the health; be it gut, skin Microbiome, and then mention the need of virome in Danish gut cohorts.

The reference in line 62 refers to **gut bacteria** in health and disease, not (gut) viruses. We expanded this section in the introduction and included reference to studies describing the gut virome in several diseases.

- “Both structural and functional imbalances of the gut bacteria, called dysbiosis, have been associated with diseases such as obesity (Gerard et al. 2016), diabetes (Gurung et al 2020, Zheng et al. 2018), inflammatory bowel disease (IBD) (Nishida et al. 2017), cancer (Cheng et al. 2020) and neurological diseases (Ma et al. 2019).” at lines 84-86.
- “At the same time, research on human gut viruses, collectively called gut virobiota, is still in its infancy (Garcia-Lopez et al. 2019), although

recent studies have demonstrated associations with IBD (Liang et al. 2020, Clooney et al. 2020), diabetes (Chen et al. 2021, Vehik et al. 2019) liver disease (Lang et al. 2020, Jiang et al. 2020) and cancer (Nakatsu et al. 2018)." at lines 86-88.

- "a population in which gut viromes have not been characterized before" at line 145-146.

II) Ln. 63, 69, 73 and 74, Ln.81 references are missing,

Line 63 (revised version line 87): "Research on human gut viruses, collectively called gut virobiota, is still in its infancy" reference to Garcia-Lopez et al. 2019

Line 69 (revised version line 96): "shed more light on the virobiota, and their collective genomes referred to as the virome, will pave the way to unraveling complex interactions within the gut microbiota and their effect on the human host." reference to Sutton & Hill 2019

Line 73 (revised version line 110): "Yet, several significant challenges in studying human gut viromes remain." reference of Sutton & Hill 2019

Line 74 (revised version line 110-111): "Most importantly, identification of viruses from metagenomes is hampered by incomplete databases" reference to Sutton & Hill 2019

Line 81 (revised version line 118-120): "However, despite these developments, a large fraction of human gut virome studies still ends up with a significant amount of "viral dark matter"." reference to Sutton & Hill 2019

4. What do authors meant by specialized resources in the Ln.75 ? Viral specialized resources ? Please re-frame the sentence for more clarity.

We mention specialized **viral identification tools**, meaning tools specifically developed to identify sequences of viral origin, in contrast to tools that (solely) rely on similarity searches to databases such as CAT or Kraken2.

We extended line 74 – 75: "and therefore requires specialized viral identification tools e.g. X, Y, Z" with "that do not (only) rely on similarity to known viral genomes/proteins but also look at genome structure to detect viral signatures" (line 113-114)

5. Ln.82 too much usage of incompleteness, please re-frame the sentence from "major incompleteness of taxonomically classified" to "major proportion of taxonomically unclassified"

We rephrased lines 81 – 83

"Moreover, taxonomic characterization of human gut viruses is virtually impossible due to the major incompleteness of taxonomically classified viruses"

With:

"Moreover, taxonomic characterization of human gut viruses is virtually impossible due to a major proportion of viruses being taxonomically unclassified"

6. CD-Hot tool must be cited, wherever used in the manuscript.

As per request of reviewer #1 – major comment #8, Cd-Hit is no longer used for analysis in this manuscript.

7. I strongly suggest rewriting the manuscript in a concise way, so that less critical sections can be moved to supplementary information. Present manuscript is still in a very rough draft version.

Although we acknowledge your criticism, none of the other 3 reviewers or the editorial board has suggested that the manuscript was too long. Therefore, we prefer not to move additional sections to the supplementary information.

We went through the complete manuscript and revised our writing where necessary.

Metagenomics computational analysis related comments:

8. Why authors didn't prefer the traditional and well adopted methodology for metagenomic assembly using Kraken and Jellyfish for assembling metagenomes and analyzing taxonomic abundance via Phyloseq. ? Why they prefer first de-novo assembly ? If de-novo was crucial, did authors backed-up their results with the above computational workflow for reproducible?

De novo assembly (using metaSPAdes) is commonly accepted as the preferred and most optimal approach for viral metagenomic assembly (see Roux et al. 2017, Shkoporov et al. 2019, Gregory et al. 2020). Although we are not familiar with Jellyfish, we have experience with Kraken2 for annotation of sequences. However, Kraken2 is generally considered not to be suitable for identification of phages without close neighbours in databases. Phyloseq is used to analyse beta-diversity as described on line 557.

9. Please mention what was the criteria to consider reads under "contaminome" ?

We added "using bowtie2 in" very-sensitive" mode" to the methods section (line 468).

10. Ln507 Please move the last section of the methods to the supplementary methods section. Authors can concise their methods by making a supplementary method section.

See our response to comment #7.

11. some of the tools are cited anywhere (neither in the method section nor in the results/discussion part) in the manuscript.

We made sure all tools are cited throughout the manuscript.

12. Ln525 what is the rationale behind taking 70% of the length criteria taken?

A large fraction (i.e. 70%) of the genome should be covered by reads from a specific sample in order for it to be confidently called “present” in that specific sample.

Thresholds of 70 – 75% are commonly used in literature (see Gregory et al. 2020, Shkoporov et al. 2019).

13. I cant see any link for the catalog? Is it available online ?

The catalog is publicly available at <https://doi.org/10.5281/zenodo.5173012>.

September 2, 2021

Prof. Jelle Matthijnssens
KU Leuven
Department of Microbiology and Immunology, Rega Institute, Laboratory of Viral Metagenomics
Leuven 3000
Belgium

Re: mSystems00382-21R1 (A previously undescribed highly prevalent phage identified in a Danish enteric virome catalog)

Dear Prof. Jelle Matthijnssens:

It is my pleasure to inform you that your manuscript has been accepted, and I am forwarding it to the ASM Journals Department for publication. For your reference, ASM Journals' address is given below. Before it can be scheduled for publication, your manuscript will be checked by the mSystems senior production editor, Ellie Ghatineh, to make sure that all elements meet the technical requirements for publication. She will contact you if anything needs to be revised before copyediting and production can begin. Otherwise, you will be notified when your proofs are ready to be viewed.

As an open-access publication, mSystems receives no financial support from paid subscriptions and depends on authors' prompt payment of publication fees as soon as their articles are accepted. =

Publication Fees:

We recognize that the video files can become quite large, and so to avoid quality loss ASM

suggests sending the video file via <https://www.wetransfer.com/>. When you have a final version of the video and the still ready to share, please send it to Ellie Ghatineh at eghatineh@asmusa.org.

Sincerely,

Chaysavanh Manichanh
Editor, mSystems

Journals Department
Supplemental Figure 3: Accept
Supplemental Figure 2: Accept
Supplemental Material Table 4: Accept
Supplemental Material Table 3: Accept
Supplemental Material Table 2: Accept
Supplemental Figure 4: Accept
Supplemental Material Table 1: Accept
Supplemental Material Table 5: Accept
Supplemental Figure 1: Accept

mSystems00382-21 review

This paper describes a new virome database named DEVOC, used here to characterize the healthy gut virome of 91 children, adolescent and adults. This effort is currently highly needed in the community, and this paper is making a very good contribution.

The authors successfully addressed all the comments, and I believe the paper is now ready for publication.

The Authors have systematically and point by point addressed my all comments. I recommend the acceptance of the manuscript.